# WildfireSpreadTS: A dataset of multi-modal time series for wildfire spread prediction

**Sebastian Gerard, Yu Zhao, Josephine Sullivan**
KTH Royal Institute of Technology
11428 Stockholm, Sweden
{sgerard, zhao2, sullivan}@kth.se

## Abstract

We present a multi-temporal, multi-modal remote-sensing dataset for predicting how active wildfires will spread at a resolution of 24 hours. The dataset consists of 13 607 images across 607 fire events in the United States from January 2018 to October 2021. For each fire event, the dataset contains a full time series of daily observations, containing detected active fires and variables related to fuel, topography and weather conditions. The dataset is challenging due to: a) its inputs being multi-temporal, b) the high number of 23 multi-modal input channels, c) highly imbalanced labels and d) noisy labels, due to smoke, clouds, and inaccuracies in the active fire detection. The underlying complexity of the physical processes adds to these challenges. Compared to existing public datasets in this area, WILDFIRESPREADTS allows for multi-temporal modeling of spreading wildfires, due to its time series structure. Furthermore, we provide additional input modalities and a high spatial resolution of 375m for the active fire maps. We publish this dataset to encourage further research on this important task with multi-temporal, noise-resistant or generative methods, uncertainty estimation or advanced optimization techniques that deal with the high-dimensional input space.

## 1 Introduction

With progressing climate change, the risk and severity of wildfires is expected to increase Shukla et al., 2019. Current active fire products Wooster et al., 2021, that monitor such wildfires, use satellite-based observations to detect the current locations of wildfires, but do not predict their future spread. Regarding the use of modern deep learning in this field, much of the existing research focuses on detection, too. This includes work on early detection and continuous mapping of active fires Zhao et al., 2022, as well as the detection of burned areas Ban et al., 2020. To go beyond detection and predict future fires, several datasets Sayad et al., 2019; Tavakkoli Piralilou et al., 2022; Prapas et al., 2021; Prapas et al., 2022; Kondylatos et al., 2022 have recently been published to estimate fire risk.

However, to fight an ongoing fire, predictions should be conditioned on the already existing fire, and have a high temporal resolution. While some research exists in this area (see section 3), the topic of predicting the future spread of active fires is still under-explored and most existing papers do not publish the data they use. This lack of public datasets makes it harder for researchers who are unfamiliar with the acquisition of remote sensing data to contribute their knowledge and skills. By releasing WILDFIRESPREADTS, we want to help bridge this gap.

For a fire to burn, three components are required: fuel, oxygen, and heat. How quickly and in which direction the fire spreads is governed by fuel conditions, wind, and topography. Relevant fuel conditions include the type and amount of fuel, as well as its humidity. This dataset is a combination of existing data products that capture different aspects of all of these features, as well as the active fire masks indicating the fire's location. To determine the location of active fire, we use the VIIRS active

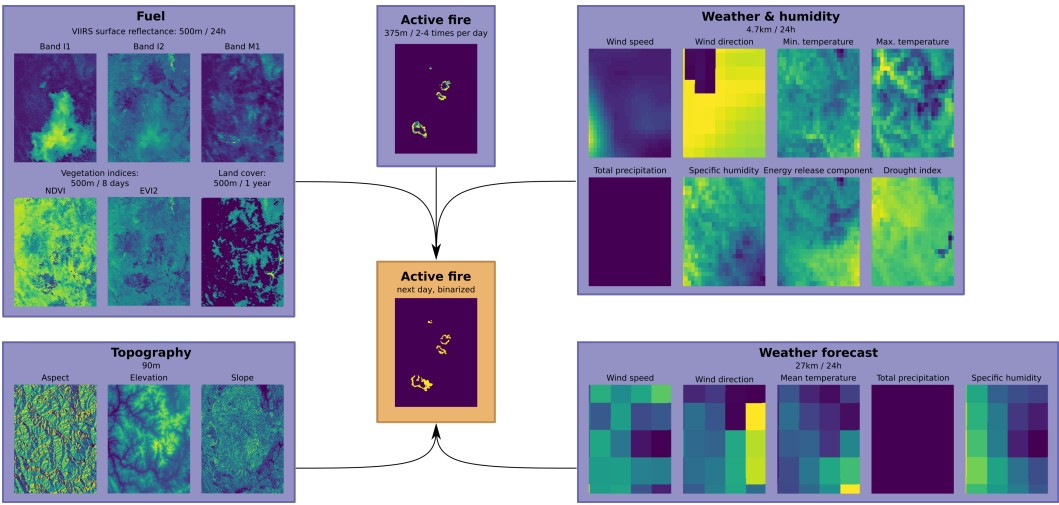

Figure 1: **Dataset overview:** The dataset's central task is to predict the next day's binarized active fire map (center box), based on one or more days of input observations (outer boxes). The input features for a day are the day's active fire map and features related to fuel, topology and weather. All features were resampled to the spatial resolution of the active fire maps of 375m, and a temporal resolution of 24h. Differences in the original spatial resolutions are clearly visible, for example in the weather forecast data, which has a resolution of roughly 27km.

fire product Schroeder et al., 2014, which detects fires based on satellite observations. The central task that the combined dataset represents is to predict the output of the VIIRS active fire product, as a proxy for predicting the actual ground truth fire spread.

WILDFIRESPREADTS is organized as a collection of image time series, with a consistent temporal and spatial resolution. This structure makes it simple to process with existing architectures, for example architectures designed for video data. We provide baseline results for predicting the next day's binarized active fire map, based on either one day (mono-temporal) or five days of observations (multi-temporal). The time series structure enables users to also construct different tasks, like predicting the wildfire spread several days in advance, or predicting the duration of a wildfire.

The only dataset we are aware of that represents wildfire spread predictions at a similar resolution is NEXTDAYWILDFIRESPREAD Huot et al., 2022. It proposes to predict the next day's MODIS-based active fire detections, based on mono-temporal observations at a resolution of 1km. Our work is heavily inspired by Huot et al., 2022. We discuss the differences to this dataset in section 3.

**Structure of this paper** In section 2, we describe the dataset in detail, including dataset statistics and visualizations of dataset features. In section 3, we describe related work in the field of wildfire spread prediction. We apply several deep learning architectures, described in section 4, to establish baseline performances, which are reported in section 5. Finally, we suggest several directions of research that might improve predictions in section 6 and indicate how to access the data in section 7.

## 2   Dataset

The WILDFIRESPREADTS dataset consists of 607 wildfire events in the contiguous United States from 2018 to 2021. The choice to focus on the United States was made solely based on the availability of data products, primarily the weather data from GRIDMET (the Gridded Surface Meteorological dataset). While we focus on the US, we hope this dataset will be useful in the development of new methods for deep learning-based wildfire spread prediction. Transferring such new methods to other regions of the world will then be an important task beyond WILDFIRESPREADTS. This transfer will require addressing the additional challenges of generalization, and adapting to data sources with potentially different spatial and temporal resolutions.

Since the source data products we use differ in spatial and temporal resolutions, we resample them to the 375m resolution of the active fire maps. This is done by Google Earth Engine internally, using

bilinear resampling. In the VIIRS active fire product, several active fire detections are possible within one day. We aggregate them in 24h windows starting at midnight and resample all other data to this temporal resolution. Figure 1 shows an overview over all features available for each day, grouped by feature category, as well as the original resolutions.

The following section provides detailed descriptions of how we created the dataset from the different original data sources. Afterwards, subsection 2.2 presents summary statistics. Finally, subsection 2.3 illustrates how these statistics translate into real data with an illustrative example fire.

## 2.1 Detailed descriptions

This section provides details about the source data products, how and why we include them. The data processing described here is performed on the Google Earth Engine Gorelick et al., 2017 (GEE) platform. All of the used datasets are available in GEE, except for the VIIRS active fire product.

### 2.1.1 Selecting fire events from GlobFire

The core component of this dataset are the active fire maps, generated from the VIIRS Active Fire (AF) Oliva et al., 2015; Schroeder et al., 2014 product. This product only consists of individual coordinates of detected active fires, but does not combine them into separate fire events. To ensure that our dataset consists of fire events of relevant size, we consulted the GlobFire Artés et al., 2019 dataset. The GlobFire dataset provides the burned area polygons, and the start and end date of a large number of wildfire events. We query the dataset for all fires larger than 1000 hectares in the years 2018 to 2021. For each of these, we extract features in a rectangle of side length 1° in latitude and longitude around the center point. Due to the way GEE internally stores the data, the resulting images vary in size. To perform multi-temporal predictions with a fixed-size input window, we also require observations from before the first day of the fire event. Therefore, we add four days before the beginning of the fire event in GlobFire. We also add four additional days at the end. We discarded any fire event which did not contain any active fire detections within the dates indicated by GlobFire and a select few events for data format issues. All discarded events were relatively equally distributed across the years. In total, we retained 607 fire events and discarded 948.

The cause of the large number of events without active fire is likely the mismatch between the 500m MODIS burned area product MCD64A1 Giglio et al., 2021, which is the basis for GlobFire, and the 375m VIIRS AF product VNP14IMG. This mismatch includes: a) the difference between burned area and active fire, b) conditions leading to false positives at 500m, but not at 375m and c) the temporal gap between when MODIS and VIIRS satellites observe the area.

### 2.1.2 Active fire maps

The VIIRS Active Fire (AF)Oliva et al., 2015; Schroeder et al., 2014 product is based on satellite observations by the VIIRS sensor onboarding the Suomi National Polar-Orbiting Partnership (S-NPP) satellite. AF detection mainly relies on the mid-infrared band I4 with wavelength 3.55-3.93 $\mu$ m. It has 375m spatial resolution with two to four detections per day, depending on the latitude. We aggregate all the detections within a daily 24h window into one active fire mask. If a pixel is detected as active fire, its value is the latest time at which it was detected to be active. In Figure 1, the different colors in the AF map indicate different detection times. The AF product also associates a three-level confidence rating with each detection. We remove all low-confidence detections.

### 2.1.3 Detecting burned area, smoke and live fuel

The VIIRS Surface Reflectance product (VNP09GA) Vermote et al., 2016 is based on the same satellite sensor as the AF product. It provides one image per day with spatial resolution between 500 meters (Band I1-I3) and 1000 meters (Band M1-M11). We include bands I1, I2 and M11 because of their ability to distinguish healthy vegetation from burned areas and to detect clouds and smoke.

The widely-used vegetation indices NDVI and EVI2 are computed in the VNP13A1 VIIRS Vegetation Indices Didan et al., 2018 product, based on a 16-day windows of these observations. The product has a spatial resolution of 500m and a temporal resolution of 8 days. These indices provide more long-term information about living fuel than the surface reflectance data.

Table 1: **Data points per year:** 2019 has unusually few fire events, images, and active fire detections within these images.

|  |  | Image summary | |
| --- | --- | --- | --- |
| Year | # fires | # imgs | % w/o detect. |
| 2018 | 176 | 3773 | 39.8% |
| 2019 | 74 | 1425 | 48.9% |
| 2020 | 201 | 4292 | 40.8% |
| 2021 | 156 | 4117 | 35.1% |

Table 2: **The persistence baseline** predicts that the fire stays the same. We compare: a) using all fire detections of the day with b) using only the latest detections, on average precision (AP) and F1 score.

|  | all detections | | last detection | |
| --- | --- | --- | --- | --- |
| Year | AP | F1 | AP | F1 |
| 2018 | 0.187 | 0.432 | 0.121 | 0.317 |
| 2019 | 0.105 | 0.323 | 0.086 | 0.279 |
| 2020 | 0.194 | 0.439 | 0.144 | 0.345 |
| 2021 | 0.287 | 0.535 | 0.189 | 0.392 |
| mean | 0.193 | 0.432 | 0.135 | 0.333 |

### 2.1.4 Weather data

The Gridded Surface Meteorological Dataset (GRIDMET)Abatzoglou, 2013 consists of daily meteorological data. We use its minimum and maximum surface temperature, total precipitation, wind speed and direction, specific humidity and the Palmer Drought Severity Index (PDSI) data. It also contains the energy release component, which relates to the energy released during combustion, and fuel moisture Bradshaw et al., 1984. The spatial resolution of the dataset is about 4.6 km. All of these features are the same as in Huot et al., 2022.

### 2.1.5 Weather forecasts

The Global Forecast System (GFS)Clough et al., 2005 provides hourly weather forecasts at a spatial resolution of 27.83km. The forecast features are similar to those provided by GRIDMET, except that GFS forecasts the average temperature (instead of min/max) and the wind as wind speed in u and v direction. We recompute the wind into overall wind speed and wind direction, to be consistent with GRIDMET. GFS provides new hourly predictions four times per day. To harmonize these frequent predictions with our 24h aggregation window, we decide to use the 24 hourly predictions made by GFS at midnight of the new day and aggregate all of them into one prediction.

### 2.1.6 Land cover and topography data

To understand the fuel type and topography of the surface, land cover and elevation data are included. For land cover data, the MODIS Land Cover Type Yearly Global product (MCD12Q1.061)Sulla-Menashe et al., 2019; Friedl et al., 2022 offers 500m spatial resolution. We use the land cover types defined by the Annual International Geosphere-Biosphere Programme (IGBP) classification criteria. For topography data, we use the NASA SRTM Digital Elevation dataset JPL, 2013 to provide elevation, and derived therefrom, slope and aspect of the ground surface. These are relevant because with rising elevation, the oxygen level changes, fires move quicker up a slope, but slower down a slope, and the aspect indicates in which direction this effect takes place.

## 2.2 Dataset statistics

The dataset consists of 607 fire events in the years 2018 to 2021 in the contiguous United States, with a total of 13607 daily images. Due to Google Earth Engine's internal processing, images vary in size, with sizes between $(304 \times 207)$ (smallest) and $356 \times 308$ (largest). The dataset is highly imbalanced, with about 0.1% of pixels representing active fire, and the rest representing no active fire.

Table 1 shows the aggregate number of fire events and images per year, as well as how many days contain fire detections at all. Figure 2 and Figure 3 visualize the corresponding distributions over all years. The low number of 74 fire events for the year 2019 is caused by few candidate events in the GlobFire dataset, as well as containing many events that we had to discard. Even though we queried the GlobFire dataset for the whole United States, only fire events in the western part of the country are part of the dataset. This is because most discarded events happened to be in central US states or Florida. In the supplementary material, we visualize the locations of included fire events on a map.

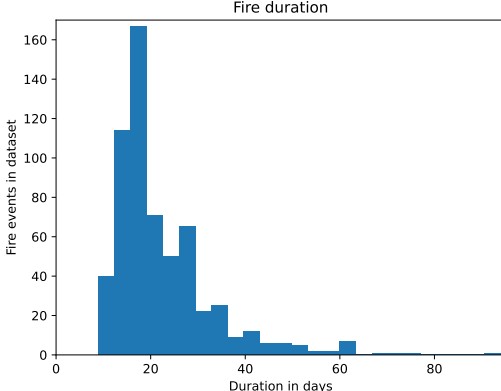

Figure 2: **Some fires burn longer than others**. The distribution of fire durations is concentrated at 14 to 20 days with a long tail that ends at 94 days. This includes the four buffer days, each before and after the dates of each fire event.

Figure 3: **Not all images contain active fires.** All fire events have at least one day with active fire detections, but many events have very few of such days. The histogram includes the four buffer days, each before and after the dates of each fire event.

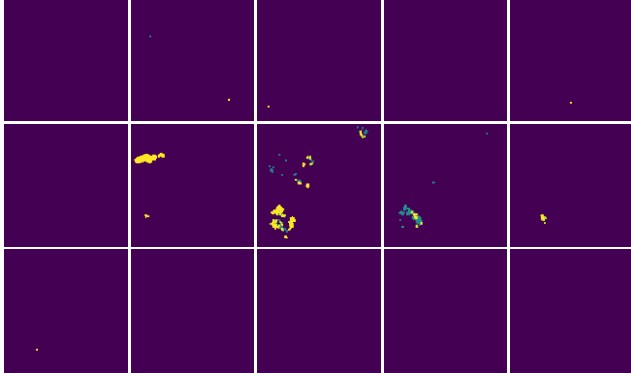

Figure 4: **Example active fire time series:** The image shows an active fire time series from 2020. Colors indicate different detection times on the same day, acquired at different overpass times by the VIIRS satellites. The time series was cropped to the $128 \times 128$ area that contains the most fire.

## 2.3 Example active fire time series

Figure 4 shows an example time series from 2020, highlighting key characteristics and challenges of the dataset. The first challenge is the difficulty of predicting the physical progression of fires. Many fires consist only of a few AF detections without any larger fire clusters (first row). Some fires then suddenly ignite into a larger fire and progress in a way that is hard to predict (second row). Other fires grow and progress more slowly and predictably. The second challenge relates to labelling noise. In Figure 4, we see the four buffer days at the start and end of the sequence contain AF pixels even though GlobFire reported the fire was not active during these times. In contrast, within the official fire dates there are time-steps containing no AF detections.

## 3 Related work

In this section, we first give a broad overview of existing work in wildfire spread prediction, and then delineate how our work differs from the closest existing one.

**Simulation software:** In practice, wildfire spread prediction is performed with simulation models like FARSITE Finney, 1998 or Prometheus Tymstra et al., 2010. Such simulators have been used to generate training data, and train neural networks to approximate the spread predictions made by the simulators Hodges et al., 2019; Burge et al., 2022; Bolt et al., 2022. A big advantage of these

methods is that they can generate data at high resolutions, for example (30 meters × 15-30 minutes) in Burge et al., 2022; Bolt et al., 2022. However, such data is not readily available for real-world situations and the simulator-based modeling might contain errors that prevent them from working optimally on actual satellite observations. Bolt et al. therefore suggest as future work to pretrain models on simulated data, and then transfer them to real-world observations.

**Tabular data:**, Singla et al. published WILDFIREDB, a wildfire spread prediction dataset in the form of tabular data, predicting whether a burning cell will spread its fire to a neighboring cell. In comparison, our dataset adds both a spatial and temporal dimension, enabling methods to be used that explicitly model these relationships, e.g. computer vision methods.

**Computer-vision:** Radke et al.; Khennou et al. and Subramanian et al. use Landsat 7 or 8 images, that have a high spatial resolution of 30m, but only provide an image every 8 days, assuming no cloud- or smoke-cover. In contrast, we use daily observations. First results in Radke et al., 2019; Ntinas et al., 2017; Ross, 2021 indicate that deep learning methods can sometimes beat FARSITE, motivating further research on learning directly from observational data.

**Daily multi-temporal input:** For the special case of Canadian peat fires, Bali et al. publish a dataset of predicting the burned area. Huot et al. investigate the problem of using one day's observations for predicting either the next day's fire or the aggregated fires over the following week, as well as using a week-long time series of observations to predict the aggregated burn area. The multi-temporal dataset used in Huot et al., 2021 has not been made available to the public, to the best of our knowledge. In Huot et al., 2022, a corresponding *mono*-temporal dataset is published, called NEXTDAYWILDFIRESPREAD. It focuses on the task of predicting the next day's wildfire spread, based on a single day of input data. A variant of the dataset has been published in the context of a disaster response hackathon Jeffrey et al., 2023. It uses the same VIIRS active fire product that we use, but is otherwise similar to NEXTDAYWILDFIRESPREAD.

## 3.1 Comparison to NextDayWildfireSpread

Since Huot et al. already published a *mono*-temporal dataset for wildfire spread prediction, we adopted many of their choices when creating this dataset. In this section, we discuss the important differences between their datasets and ours.

**Time series structure** NEXTDAYWILDFIRESPREAD consists of input-output pairs of single-day observations and targets. In contrast, WILDFIRESPREADTS consists of 607 multi-day time series, each associated with an individual fire event. This structure allows the construction of not only multi-day inputs, but also custom task definitions, like predicting active fire one or seven days in advance, or predicting the duration of a fire. This extension of the data into the temporal dimension, which enables multi-temporal modeling, is the core contribution of our paper.

**More recent observations** To describe the state of the vegetation, Huot et al. use the vegetation index NDVI, which is updated every 8 days. We add to this the daily VIIRS reflectance observations, to receive more frequent updates about vegetation conditions. This seems especially important in the context of longer-lasting fires, where using data that is seven days old might lead to sub-par results. Together with moving to a multi-temporal setup, the VIIRS observations provide the model with more up-to-date information, thereby potentially improving predictions.

**Additional input features** We add weather forecast data, which seems vital to predict whether a fire is going to be extinguished by heavy rain, or whether the flames will be fanned into a certain direction by strong winds. Since the direction and speed of fire spread is also impacted by the local topography, we add aspect and slope as features. We add land cover classes to differentiate fuel types and detect the presence of fire-impeding land covers, like barren or built-up land, or water bodies.

**Fire masks** NEXTDAYWILDFIRESPREAD is based on MODIS observations with a spatial resolution of 1km. All features are resampled to this resolution. As we use VIIRS active fire labels, we resample all data to 375m, instead. This increased resolution in our dataset leads to increased resolutions for topographical features, vegetation indices, surface reflectance and land cover, which might positively affect predictive performance. We also provide more fine-grained temporal information for the input active fire masks. Since VIIRS active fire detections occur 2-4 times per day, we indicate for each pixel the hour at which it was last detected as burning. In section 5, we predict the next day's fire as a

binary label. However, the temporal information included in the active fire masks could also be used to predict sub-daily wildfire spread, by predicting the detection times.

Due to the already large file size, caused by the number of additional features, we currently only use data from 2018 to 2021, while Huot et al. use data from 2012 to 2020. We also do not include proxies for anthropogenic activity, since these features are mostly important for predicting the risk of ignition, as far as we are aware, while we care about the situations in which ignition has already occurred.

## 4 Method

To establish performance baselines, we apply several widely-used architectures to our dataset.

**Logistic Regression** Huot et al. use a logistic regression with a $3 \times 3$ input window as a simple baseline model. We implement this architecture as as a single convolution with a $3 \times 3$ kernel. We only use mono-temporal data for the logistic regression.

**U-Net** We use a U-Net Ronneberger et al., 2015, based on a ResNet18 He et al., 2016 architecture, for both mono- and multi-temporal data. Since it has no direct way to handle multi-temporal data, we concatenate all time steps in the channel dimension.

**ConvLSTM** The ConvLSTM Shi et al., 2015 is a recurrent neural network architecture based on convolutional layers and able to process multi-temporal image data. We use a single ConvLSTM block, followed by a final convolution that produces the segmentation map.

**U-Net + temporal component** Several architectures exist that combine the U-Net architecture with a temporal component. We use a version called UTAE Garnot et al., 2022 that applies a simplified multi-head self-attention across the temporal dimension at the bottleneck. The generated attention map is then upscaled and applied to all skip-connections between encoder and decoder.

## 5 Experiments

**Software & Hardware** All of our models, experiments, training procedures and data handling are implemented in Python using PyTorch Paszke et al., 2019 and Lightning Falcon et al., 2019, and logged via Weights and Biases Biewald, 2020. We used NVIDIA A40 GPUs situated in the NAISS cluster Alvis (see acknowledgements). The experiments reported here, including hyperparameter searches, took 1278 hours. With debugging, re-runs due to errors etc., computations took about four times as long.

### 5.1 Model and training hyperparameters

Unless mentioned otherwise, we use the default hyperparameters of the respective implementations. Loss functions and learning rates are chosen via coarse grid searches. Please see the supplementary material for details. We always use the PyTorch implementation of the AdamW optimizer Loshchilov et al., 2019 with the default parameters $\beta_1 = 0.9, \beta_2 = 0.999, \lambda = 0.01$. We keep the model with the highest validation loss, evaluated after each training epoch and train for a total of 10k steps.

For the logistic regression, we use a Dice loss with learning rate 0.1. For the U-net, we use the SMP implementation Iakubovskii with a ResNet18 backbone and train it with a Dice loss with learning rate 0.001. For ConvLSTM and UTAE, we use the implementations VSainteuf, 2023 by the UTAE Garnot et al., 2022 authors. For ConvLSTM, we use a Jaccard loss with learning rate 0.01. For UTAE, we use a weighted cross-entropy and learning rate 0.01. The weight for the fire class is the inverse relative frequency of the fire class in the used training set.

### 5.2 Dataset preparation

We split the data into train/val/test by using two years for the train set and one each for validation and test set. For all experiments, we perform a full 12-fold cross-validation over all possible permutations of years assigned to train/val/test set. This is necessary, because of the large differences between the yearly data distributions (see Table 1 and Table 2, more details in the supplementary material).

For fair evaluation, we need to ensure that the test data is the same across all experiments. However, the number of data points per fire depends on the size of the temporal window $w_t$, i.e. the number of observed days used as input. E.g. for $w_t = 5$, we would use the first five days as input, and the binary fire mask of the sixth day as the first target, while for $w_t = 1$, we would use the second day as our first target, leading to differences in the test set. The reason for this discrepancy is that we only use real data as input to our models, since it is realistic to have access to past data, instead of padding the input with empty frames. In our experiments, we use a maximum $w_t = 5$, meaning that the first target used in evaluation belongs to day six. For evaluation on the test set to be fair across different values of $w_t$, we have to leave out initial data points in experiments with $w_t < 5$, to ensure that all testing starts on the target of day six, for each fire. During training, we instead use all of the data. For details on preprocessing and augmentations, please see the supplementary material.

To evaluate models, we use the test set average precision (AP). It summarizes the precision-recall curve in a single number and is preferable to metrics like AUROC in cases of imbalanced datasets, like WILDFIRESPREADTS. Compared to the F1 score, it considers all possible thresholds instead of just one. For the persistence baseline, we still provide the F1 score, to give readers who are more familiar with this metric an intuition of how good the corresponding AP scores are.

## 5.3 Fire persistence baseline

As a simple baseline, we use fire persistence, as in Huot et al., 2022. It consists of predicting the same active fire for tomorrow as was observed today. Table 2 shows that the performance of this baseline varies strongly between the different years. Together with the lower number of fires for 2019, this indicates that the difficulty of predicting wildfire spread might vary accordingly.

Since our data includes the time at which each active fire pixel was last detected, we can construct a more fine-grained variant of this baseline, using only pixels from the last detection of the day. If a pixel was detected as active fire early in the day, but not in the last detection of the day, it might have stopped burning. The results in Table 2, however, show a worse performance for this fine-grained baseline. The causes could be: 1. The later detections did not cover the exact same area as the earlier ones, thereby leaving out some burning pixels that were out of view. 2. The detections are noisy and might fail to recognise some of the pixels as burning. This could be caused by different view angles between observations, or temporal variations in fire intensity or cloud/smoke cover.

## 5.4 Ablation studies

To investigate how useful different features are, we combine the active fire masks with different feature groups. We train the U-Net on the resulting feature set, varying the numbers of leading days of observations. In preliminary experiments, we found that the U-Net is very easy to optimize, even though it does not explicitly model the temporal dimension. Explicit spatio-temporal models proved difficult to optimize in some cases, which is why we use the U-Net for the ablation studies.

**Ablation: Vegetation** Table 3 compares the common vegetation indices NDVI and EVI2 with the daily VIIRS reflectance product that we add in this dataset. For one and two days, VIIRS performs better than NDVI, which was the best one-feature solution in Huot et al., 2022, Table V. For three to five days of input observations, VIIRS, NDVI and EVI2 perform similarly, although the exact ranking fluctuates. This strong performance of VIIRS on shorter time series might be caused by the higher temporal resolution, getting daily updates instead of every eight days, and VIIRS bands also offering information about the presence of clouds and smoke. Observing multiple leading days also improves performance for all vegetation features, validating the addition of the temporal dimension.

**Ablation: Non-vegetation** In Table 4, we test all non-vegetation features in the same setting. We see that topography features perform best, though still worse than the VIIRS features. Again, increasing the number of observed leading days increases the model's performance. Notably this is also true for static features like topography.This indicates that the additional active fire observations contribute some of the improvement, possibly by indicating already burnt areas. Each combination of fire mask and individual feature group and outperforms the persistence baseline that reaches a mean AP of 0.189. Even the worst performing result in this ablation study, ERC & drought, with a single input observation, has a mean AP of 0.291. We also observe that performance can decrease when adding more time steps, even though the model could just learn to ignore the additional data and retain its performance. This could be an issue of optimization difficulty in high-dimensional feature spaces.

Table 3: **Ablation: Vegetation** We compare EVI2, NDVI and VIIRS reflectance as sources for vegetation data, combined with fire masks. The models predict the wildfire spread 24h ahead, based on one to five days of observations. The performance displayed is the mean test set **average precision ± the standard deviation**, computed in a 12-fold cross-validation over years. Higher is better.

| Input days | Features | | |
|:---:|:---:|:---:|:---:|
| | EVI2 | NDVI | VIIRS |
| 1 | 0.302 ± 0.096 | 0.298 ± 0.091 | **0.314 ± 0.093** |
| 2 | 0.313 ± 0.091 | 0.314 ± 0.085 | **0.319 ± 0.087** |
| 3 | **0.321 ± 0.092** | 0.318 ± 0.087 | 0.320 ± 0.085 |
| 4 | 0.319 ± 0.084 | **0.325 ± 0.087** | 0.323 ± 0.091 |
| 5 | 0.318 ± 0.085 | **0.319 ± 0.080** | **0.319 ± 0.087** |

Table 4: **Ablation: Non-vegetation** We compare all non-vegetation features, individually combined with the fire masks as input features. The models predict the wildfire spread 24h ahead, based on one to five days of observations. The performance displayed is the mean test set **average precision ± the standard deviation**, computed in a 12-fold cross-validation over years. Higher is better.

| Input days | Features | | | | |
|:---:|:---:|:---:|:---:|:---:|:---:|
| | ERC, drought | Landcover | Topography | Weather | Weather forecast |
| 1 | 0.291 ± 0.088 | 0.301 ± 0.094 | **0.306 ± 0.087** | 0.295 ± 0.091 | 0.296 ± 0.098 |
| 2 | 0.300 ± 0.088 | 0.311 ± 0.087 | **0.315 ± 0.087** | 0.308 ± 0.089 | 0.296 ± 0.094 |
| 3 | 0.303 ± 0.096 | 0.314 ± 0.084 | **0.317 ± 0.082** | 0.316 ± 0.087 | 0.292 ± 0.103 |
| 4 | 0.308 ± 0.087 | 0.307 ± 0.084 | **0.321 ± 0.089** | 0.313 ± 0.093 | 0.289 ± 0.094 |
| 5 | 0.313 ± 0.086 | **0.318 ± 0.086** | 0.317 ± 0.080 | 0.308 ± 0.091 | 0.293 ± 0.096 |

Table 5: **Baseline results** The models predict the wildfire spread 24h ahead, based on one or five days of observations and different input features. *Multi* includes vegetation, land cover, topography and weather features. The performance displayed is the **mean test set average precision ± the standard deviation**, computed in a 12-fold cross-validation over years. Higher is better.

| Model | Input days | Features | | | Parameters (All) |
|:---|:---:|:---:|:---:|:---:|:---:|
| | | Vegetation | Multi | All | |
| Persistence | 1 | 0.193 ± 0.065 | 0.193 ± 0.065 | 0.193 ± 0.065 | 0 |
| Log. Regression | 1 | 0.279 ± 0.092 | 0.288 ± 0.091 | 0.286 ± 0.092 | 361 |
| Res18 U-Net | 1 | 0.328 ± 0.090 | 0.341 ± 0.085 | **0.341 ± 0.086** | 14.4M |
| Res18 U-Net | 5 | 0.333 ± 0.079 | 0.344 ± 0.076 | 0.325 ± 0.108 | 14.7M |
| ConvLSTM | 5 | 0.306 ± 0.082 | 0.310 ± 0.085 | 0.292 ± 0.094 | 240K |
| UTAE | 5 | **0.372 ± 0.088** | **0.350 ± 0.113** | 0.321 ± 0.135 | 1.1M |

**Ablation: Combining features** To assess the cumulative contribution of these features, we sequentially combine features. Testing all feature combinations for all time windows is computationally prohibitive. Instead, we sort feature groups in descending order w.r.t. performance on the ablation experiments reported in Table 4, and then add them cumulatively one by one. To reduce experiments, we group all vegetation features together and only compare time windows of one and five days. Table 2 in the supplementary material shows the results in detail. The performance does not always improve with more features or input days. Instead, more features or days sometimes lead to worse results. This is an issue of optimization, possibly made more difficult by the higher input dimensionality, since the model could theoretically just ignore the additional inputs. The best performance is achieved by models using vegetation, land cover, topography and weather data, with five days of input observations. Adding weather forecast, ERC and drought results in worse performance.

### 5.5 Baseline results

To investigate how different baseline models perform on the dataset, we train them on: a) the best one-feature setting from Table 3 and Table 4, b) the best multi-feature setting (Table 2 in the supplementary material) and c) all available features. The results are displayed in Table 5.

The best result by far is achieved by the spatio-temporal UTAE method, with five input days, using only the vegetation features. This validates moving from mono-temporal methods, that only model spatial interactions, to multi-temporal methods, that model spatial and temporal interactions. The UTAE achieves a maximum AP of 0.372 on the vegetation features (plus the fire masks which are always included). Adding more features reduces the performance to 0.350 for the multi-feature setting and to 0.321 when using all features in the dataset. This hints at optimization difficulties in high-dimensional feature space, similar to what we observed in the ablation studies.

The next-best model is the U-Net with a maximum AP of 0.341 for the mono-temporal and 0.344 for the multi-temporal settings. Notably, the UTAE has about 13 times fewer parameters than the U-Net, but outperforms it by up to 3.9 pp. The U-net does outperform the ConvLSTM, the second spatio-temporal model we tested, which reaches a maximum AP of 0.310. However, the ConvLSTM also has 61 times fewer parameters than the U-Net. The logistic regression provides a strong baseline, achieving a maximum AP of 0.288. All of the parametric models very clearly beat the parameter-free persistence baseline that has an AP of 0.193, showing that they do indeed learn meaningful features.

## 6  Opportunities

In this paper, we introduced `WildfireSpreadTS`, a dataset dedicated to predicting the spread of wildfires based on time-series data. We believe that a wide variety of methods are promising avenues to improve the state of the art of predictions on this important problem.

**Optimization** We have shown that parameter optimization with this high-dimensional, multi-modal dataset can be difficult. While we used standard techniques, more sophisticated methods for optimization with high-dimensional data could improve upon these issues.

**Self-supervised pretraining** Self-supervised learning (SSL) has proven to be useful for satellite observations Ayush et al., 2021; Gerard et al., 2022; Cong et al., 2023. Additionally, Kang et al., 2021 found that SSL pre-training is beneficial for class-imbalanced datasets. Applying this pretraining could alleviate the optimization difficulties that we encountered.

**Noisy labels** Since the active fire detections are impacted by various noise sources in the acquisition process, using methods that are tolerant towards label noise Song et al., 2022; Englesson et al., 2021 could improve the robustness of results. Alternatively, uncertainty estimation Gawlikowski et al., 2022 or generative modeling could be approaches to deal with the fact that the process of fire spreading likely contains some stochasticity at the resolution that we are working with.

**Separating static and dynamic features** While this dataset is structured such that every day provides all features, some of them do not actually change on a daily basis. Treating these static and dynamic features separately, as done by Eddin et al., could facilitate the optimization problem.

## 7  Access to the dataset

The most up-to-date information on the dataset, as well as the code used in this paper, can be found at `https://github.com/SebastianGer/WildfireSpreadTS`. This includes a PyTorch dataset class, and a Lightning data module. Our code is available under the MIT license.

The full dataset Gerard et al., 2023 of about 50GB is available as a collection of GeoTIFF files under the CC-BY-4.0 license at `https://doi.org/10.5281/zenodo.8006177`. For training, we recommend converting the dataset to HDF5. The corresponding code is available in our repository.

## Acknowledgments and Disclosure of Funding

This work is funded by Digital Futures in the project EO-AI4GlobalChange. The computations were enabled by resources provided by the National Academic Infrastructure for Supercomputing in Sweden (NAISS) at C3SE partially funded by the Swedish Research Council through grant agreement no. 2022-06725.

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
