# Supplementary material

## 1 Dataset documentation

In this section, we follow the Datasheets for Datasets framework Gebru et al., 2020 to document the dataset and its creation process. We excluded the questions relating to individual people, since those do not apply to this dataset.

### 1.1 Motivation

- **For what purpose was the dataset created?**

  The purpose of the dataset is to enable researchers to perform multi-temporal wildfire spread prediction. While there is existing research on this topic, to the best of our knowledge, no dataset has been publicly released, that allows this for regular wildfires (i.e. not the special case of peatland fires treated in Bali et al.). A similar dataset exists for mono-temporal wildfire spread prediction Huot et al., 2022. Our dataset moves from the mono-temporal to the multi-temporal setting. Since the dataset is structured as a set of time series, researchers can decide for themselves how many days of observations they want to use as input, how many days in the future their prediction target is or how that target is computed from the observations.

- **Who created the dataset (e.g., which team, research group) and on behalf of which entity (e.g., company, institution, organization)?**

  The dataset was created by Sebastian Gerard, Yu Zhao and Josephine Sullivan at KTH Royal Institute of Technology.

- **Who funded the creation of the dataset?**

  This work is funded by Digital Futures in the project EO-AI4GlobalChange.

### 1.2 Composition

- **What do the instances that comprise the dataset represent (e.g., documents, photos, people, countries)?**

  Each instance is one image consisting of 23 channels.

- **How many instances are there in total (of each type, if appropriate)?**

  There are 13 607 images in total. Each image is associated with one of 607 fires that took place in the western USA.

- **Does the dataset contain all possible instances or is it a sample (not necessarily random) of instances from a larger set?**

  Please refer to the main paper for the full dataset collection process. The dataset is based on the GlobFire Artés et al., 2019 database of wildfires in the USA. We only included fire events in the years 2018 to 2021 that had a size of at least $10^7 m^2$. We discarded a large share of the resulting fires, since they had no active fire detections in the active fire dataset we used Schroeder et al., 2014. The remaining fires were all concentrated in the western part of the USA. The decision to focus on the USA for fires was solely based on easily accessible base datasets, most importantly the dataset of weather variables. See Figure 1 for the spatial distribution of wildfires in the different splits.

- **What data does each instance consist of?**

  Each instance is an image with 23 channels, which are the following: VIIRS band M11, VIIRS band I2, VIIRS band I1, NDVI, EVI2, total precipitation, wind speed, wind direction, minimum temperature, maximum temperature, energy release component, specific humidity, slope, aspect, elevation, Palmer drought severity index, landcover class, forecast total precipitation, forecast wind speed, forecast wind direction, forecast temperature, forecast specific humidity, active fire.

- **Is there a label or target associated with each instance?**

  The data is organized as a set of time series, so no label is explicitly prescribed. In our baseline experiments we binarize the next day's active fire detection and use this as the prediction target. Users are free to define different targets, if desired.

- **Is any information missing from individual instances?**

  Satellite observations can have missing data for various reasons, like cloud cover or sensor failure. The respective data shows up as missing values in the images contained in this dataset. See Table 3 for statistics.

- **Are relationships between individual instances made explicit (e.g., users' movie ratings, social network links)?**

  Images belonging to the same fire event are contained in the same folder, and named with the respective date that the image represents. The GeoTIFF images also contain the geolocation of each pixel.

- **Are there recommended data splits (e.g., training, development/validation, testing)?**

  We recommend to evaluate models using a cross-validation across years, since the data distributions vary a lot between years.

- **Are there any errors, sources of noise, or redundancies in the dataset?**

  Satellite observations are always affected by noise, e.g. due to reflections, atmospheric conditions and inaccuracies in the georegistration of measurements.

- **Is the dataset self-contained, or does it link to or otherwise rely on external resources (e.g., websites, tweets, other datasets)?**

  The dataset is self-contained.

- **Does the dataset contain data that might be considered confidential (e.g., data that is protected by legal privilege or by doctor–patient confidentiality, data that includes the content of individuals' non-public communications)?**

  No.

- **Does the dataset contain data that, if viewed directly, might be offensive, insulting, threatening, or might otherwise cause anxiety?**

  No.

## 1.3 Collection Process

- **How was the data associated with each instance acquired?**

  This dataset is based on existing datasets. We did not collect any data ourselves, but selected matching data from various existing datasets to compile this dataset. The exact acquisition mode of the original data varies between the different source data products. Please refer to the original publications (see Table 1). Most of them are based on different kinds of satellite observations.

- **What mechanisms or procedures were used to collect the data (e.g., hardware apparatuses or sensors, manual human curation, software programs, software APIs)?**

  See previous question.

- **If the dataset is a sample from a larger set, what was the sampling strategy (e.g., deterministic, probabilistic with specific sampling probabilities)?**

  Not applicable.

- **Who was involved in the data collection process (e.g., students, crowdworkers, contractors) and how were they compensated (e.g., how much were crowdworkers paid)?**

This dataset is based on existing datasets. Please refer to the original publications (see Table 1).

- **Over what timeframe was the data collected?**

  The data represents fires from 2018 to 2021. Most of the included data products collect their data on the exact day represented by the respective image. Some of them have coarser temporal resolutions (see Table 1) and incorporate data from longer time periods. The elevation data in SRTMGL1 v003JPL, 2013 was collected during an individual mission in the year 2000.

- **Were any ethical review processes conducted (e.g., by an institutional review board)?**

  No.

## 1.4 Preprocessing/cleaning/labeling

- **Was any preprocessing/cleaning/labeling of the data done (e.g., discretization or bucketing, tokenization, part-of-speech tagging, SIFT feature extraction, removal of instances, processing of missing values)?**

  We resampled all features to a spatial resolution of 375m, which is the resolution of the VIIRS active fire product. Active fire detections are originally individually tagged with a timestamp for each point detected. We aggregated all detections within 24h windows within an area derived from the GlobFire database entry. We removed active fire detections that were labeled as low confidence and kept medium and high confidence detections. The GFS wind features are originally given as wind in u and v direction. We transformed those into overall wind speed and wind direction, to be compatible with the GRIDMET features. The GFS features are originally hourly, but we aggregated them to a daily resolution. For all features that are updated less often than daily, we take the most recent value, meaning that subsequent days often contain the same values for these features.

- **Was the "raw" data saved in addition to the preprocessed/cleaned/labeled data (e.g., to support unanticipated future uses)?**

  No, but all of the source data products are freely available online.

- **Is the software that was used to preprocess/clean/label the data available?**

  In our case, this is simply the software used to download the data from GEE. It is available at `https://github.com/SebastianGer/WildfireSpreadTSCreateDataset` under the MIT license.

## 1.5 Uses

- **Has the dataset been used for any tasks already?**

  The dataset has been used for predicting the next day's wildfire spread, based on one or five days of observations. See the main paper for details.

- **Is there a repository that links to any or all papers or systems that use the dataset?**

  No.

- **What (other) tasks could the dataset be used for?**

  Since the data is organized as a collection of time series, users are free to choose how many days of observations to use as input and which data to use as targets. Possible targets would be to predict active fire N days in advance, predict the aggregate of active fire detections N days in advance (see Huot et al., 2021), or predict how long a fire is going to last.

- **Is there anything about the composition of the dataset or the way it was collected and preprocessed/cleaned/labeled that might impact future uses?**

  The dataset only contains fires in the western USA and some of the data sources used are only available in the USA. Models that perform well on this dataset might therefore not be immediately usable for other parts of the world, since the respective data might not be available.

  The dataset does not contain all fires that occurred in the respective years and areas. We used the MODIS-based GlobFire database to find candidate time spans and areas and then included the respective data, if the VIIRS active fire product had detected fire in these

areas. We also filtered out fires that were below a size of $10^7 m^2$ according to GlobFire. We included 4 days of observations before and after the time span given in GlobFire and often found that they contained active fire detections (about 3/8 days on average). Therefore, the dates given in GlobFire, that also determined which time range we used for each fire, might not be completely accurate.

Since all features were resampled to a common resolution, it might appear that features were measured at that resolution. However, the original resolutions vary between features. Resampling the data does not increase the amount of information that the original data provided.

The influence of human fire-fighting activity has not been accounted for in the dataset. A fire shrinking in size might not be the natural progression, but an effect of human intervention. We are not aware of any large-scale data on such interventions and thus did not account for them.

- **Are there tasks for which the dataset should not be used?**

  The dataset should not be used as ground truth for any kinds of statistical analysis on wildfires or how they spread. For that, the users should refer to unfiltered, unprocessed data.

## 1.6 Distribution

- **Will the dataset be distributed to third parties outside of the entity (e.g., company, institution, organization) on behalf of which the dataset was created?**

  The dataset will be made publicly available.

- **How will the dataset will be distributed (e.g., tarball on website, API, GitHub)?**

  The dataset will be published on the Zenodo platform European Organization For Nuclear Research et al., 2013, freely available under the CC BY 4.0 license.

- **When will the dataset be distributed?**

  The dataset will be published with the camera-ready version of the paper.

- **Will the dataset be distributed under a copyright or other intellectual property (IP) license, and/or under applicable terms of use (ToU)?**

  The dataset will be distributed under the CC BY 4.0 license.

- **Have any third parties imposed IP-based or other restrictions on the data associated with the instances?**

  No. The base datasets we use are either in the public domain or similarly permissively licensed, only requesting citation of the original source.

- **Do any export controls or other regulatory restrictions apply to the dataset or to individual instances?**

  No.

## 1.7 Maintenance

- **Who will be supporting/hosting/maintaining the dataset?**

  The dataset will be hosted on Zenodo.

- **How can the owner/curator/manager of the dataset be contacted (e.g., email address)?**

  The first author can be reached at sgerard@kth.se.

- **Is there an erratum?**

  No.

- **Will the dataset be updated (e.g., to correct labeling errors, add new instances, delete instances)?**

  We currently do not plan to perform regular updates. If important short-comings of the dataset are detected, or if there is enough interest, we would update the dataset on Zenodo, which allows for several versions of a dataset. Any other news will be available at `https://github.com/SebastianGer/WildfireSpreadTS`.

- **Will older versions of the dataset continue to be supported/hosted/maintained?**

  Older versions of the dataset will still be available on Zenodo.

- **If others want to extend/augment/build on/contribute to the dataset, is there a mechanism for them to do so?**

  It would be possible to either contact the first author (sgerard@kth.se) to coordinate uploading a modified version, or to make use of the creative commons license and publish the modified dataset right away.

## 2   Code

The code to re-create the dataset is available at `https://github.com/SebastianGer/WildfireSpreadTSCreateDataset`. The code to re-run the experiments, as well as the dataloader and Lightning data module, is available at `https://github.com/SebastianGer/WildfireSpreadTS`. Both repositories are licensed under the MIT license.

## 3   Discarded events

As described in the main paper, we chose fire events based on their entry in the GlobFire database. While investigating the resulting images, we found that many of the fire events did not have any active fire detections in any of the resulting days, so we discarded those events. We furthermore discarded nine events that only had active fire detections outside of the time period indicated by GlobFire and three further events for miscellaneous data format issues.

This mismatch between MODIS-based GlobFire and the VIIRS active fire product further manifests in fire being present on roughly 3 out of the 8 buffer days we added outside of the official beginning and end date, indicating that the dates might not be perfectly accurate for the higher resolution VIIRS fire product.

## 4   Spatial distribution of fires

Figure 1 shows the spatial distribution of fires in an example split into train, validation and test set, annotated with the land cover class at the center pixel of each fire. How the years are actually split into train/val/test sets in our experiments is based on the chosen dataset fold in cross-validation.

## 5   Original data sources

Table 1 details the original resolutions of each feature and which dataset it stems from. Due to the very diverse temporal and spatial resolutions, we resampled everything to a resolution of 375m / 24h, to greatly facilitate using the data. Resampling data of different resolutions is a standard procedure in remote sensing. The resampling is performed by Google Earth Engine, using bilinear interpolation.

### 5.1   Noise in the AF product

The VIIRS active fires product is subject to multiple sources of noise. As Zhao et al. say: "While these products are very useful, the existing solutions have flaws, including many false alarms due to cloud cover or buildings with high-temperature roofs." Oliva et al. mention coal power plants in South Africa as an example of anthropogenic heat sources that are detected as temperature anomalies, but do not actually represent fires. They also find problems with "high fire spread rate events that burned and extinguished in the time lapse between satellite observations". Additionally, they allude to the fact that smaller fires are harder to detect.

### 5.2   Evolution of fire pixels

Table 2 shows statistics of how active fire detections evolve over time. Some fires are rather extreme, as indicated by the max values with thousands of pixels changing from frame to frame. Most fires have changes of around 40 pixels per fire, as indicated by ignition and extinguishment at the 50th percentile. In comparison, persistence is much lower with 23 pixels as the 50th percentile. We can interpret this as fires changing more than they stay the same. Looking at the frame-wise difference, we see that a median change of $\pm$ 30 pixels is normal. Note that all of these numbers exclude

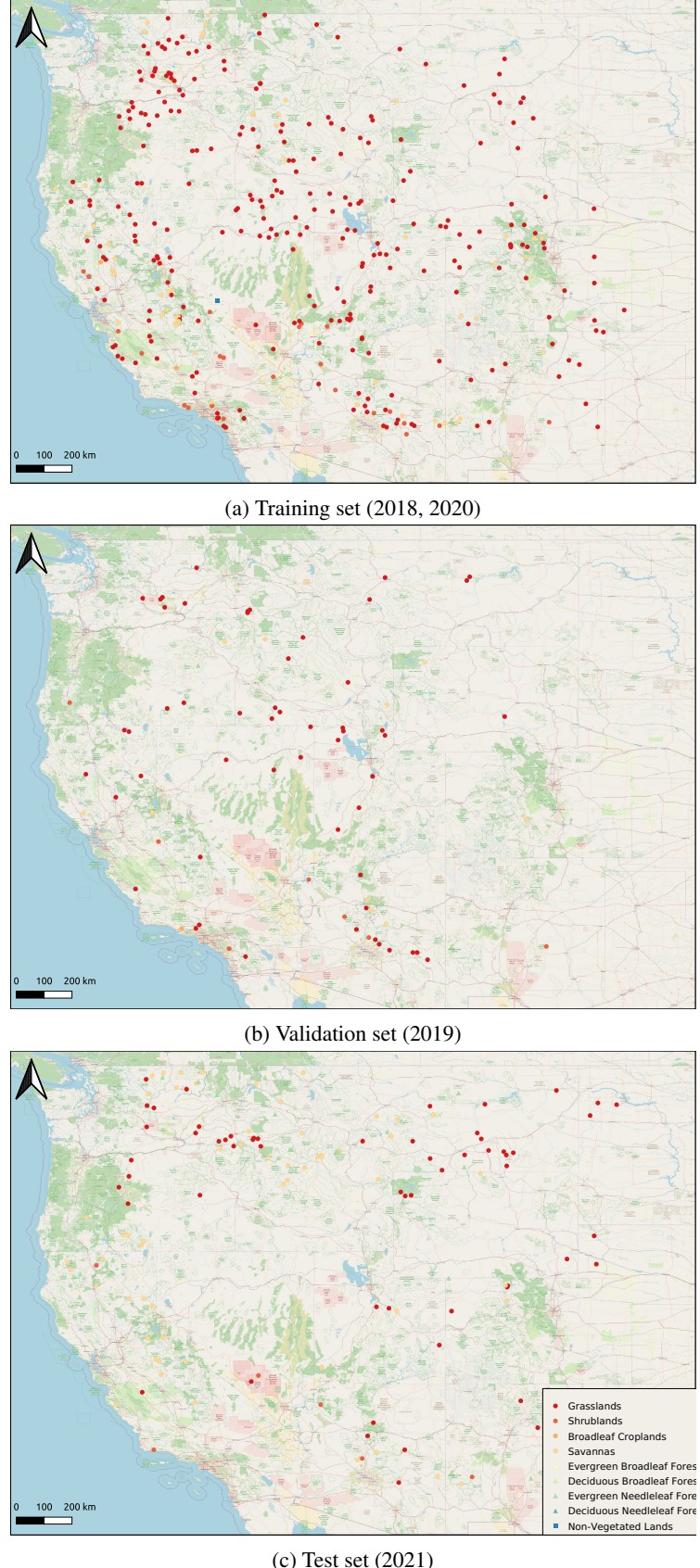

(a) Training set (2018, 2020)

(b) Validation set (2019)

(c) Test set (2021)

Figure 1: Overview of locations and land covers of fires in an example split into train/test/validation set. The legend in the test set does not cover any fires.

Table 1: Overview of data sources

| Original dataset | Feature | Resolution |
|---|---|---|
| VNP14IMG Schroeder et al., 2014 | VIIRS active fire | 375m / 24h |
| VNP09GA Vermote et al., 2016 | VIIRS surface reflectance | Bands I1, I2: 500m; Band M11: 1km / 24h |
| VNP13A1 Didan et al., 2018 | VIIRS vegetation indices | 500m / 8 days |
| SRTMGL1 v003 JPL, 2013 | Elevation (derived from this: aspect, slope) | 30m |
| MCD12Q1.061 Friedl et al., 2022 | Land cover class | 500m |
| GRIDMET Abatzoglou, 2013 | Wind, rain, temperature, humidity, drought index, energy release component | 4 638.3m / 24h |
| NOAA_GFS0P25 Clough et al., 2005 | Forecast for wind, rain, temperature, humidity | 27 830m / 1h |

Table 2: **Evolution of fire pixels** We measure how active fire pixels change between subsequent time steps. Ignition means that the a fire is detected at time step $t$ where no fire was detected at $t-1$. Persistence means that a pixel that was on fire at $t-1$ is still on fire at $t$. Extinguishment means that a burning pixel at $t-1$ does not burn anymore at $t$. Frame-wise difference indicates the difference in the number of active fire pixels between subsequent frames. We only compute ignition if the current frame has any active fires. Ignition values of 0 therefore mean that fires only persisted or were extinguished. We only compute persistence and extinguishment for frames at time $t$, when there were active fires at $t-1$. We compute the frame-wise difference only when one of the two frames contains at least one fire pixel.

|  | Ignition | Persistence | Extinguishment | Frame-wise difference |
|---|---|---|---|---|
| count | 8038.0 | 7947.0 | 7947.0 | 9185.0 |
| mean | 123.1 | 110.3 | 123.4 | 1.0 |
| std | 284.9 | 229.7 | 264.3 | 276.0 |
| min | 0.0 | 0.0 | 0.0 | -4319.0 |
| 25% | 11.0 | 2.0 | 10.0 | -32.0 |
| 50% | 44.0 | 23.0 | 42.0 | 0.0 |
| 75% | 129.0 | 119.5 | 132.0 | 30.0 |
| max | 7521.0 | 4853.0 | 4691.0 | 7423.0 |

the uninteresting transitions between frames that both do not contain any active fires. In total, we investigated 13 000 frame transitions. The count variable at the top of the table shows that for ignition, persistence and extinguishment, about 5 000 transitions were excluded because they would not have added any interesting information. That leaves about 61.5% of transitions from which these statistics are computed, or 70.7% for the frame-wise difference.

## 5.3   Missing values

Table 3 shows the rate of missing values per feature. VIIRS reflectance misses around 5% of measurements, which is less than the long-term aggregated indices NDVI and EVI2. GRIDMET features and topography are missing in about 1% of cases. Landcover type and weather forecasts are available for all pixels. Active fire shows up as missing for 99.83% of all pixels. This is due to the fact that only active fire detections are marked as known values. All pixels that are not detected as burning are marked as missing values. In preprocessing, we replace these missing values with zeros. Consequently, the amount of fire pixels in all years is 0.17%.

Table 3: **Missing values** We display the rates of missing values, aggregated over all years. The active fires feature shows 99.83% missing values, due to the fact that only active fire detections are marked as known values. All pixels that are not detected as burning are marked as missing values. In preprocessing, we replace these missing values with zeros.

| Feature | Missing values (%) |
| --- | --- |
| VIIRS band M11 | 4.95 |
| VIIRS band I2 | 4.95 |
| VIIRS band I1 | 4.95 |
| NDVI | 8.35 |
| EVI2 | 8.35 |
| Total precipitation | 1.08 |
| Wind speed | 1.08 |
| Wind direction | 1.08 |
| Minimum temperature | 1.08 |
| Maximum temperature | 1.08 |
| Energy release component | 1.08 |
| Specific humidity | 1.08 |
| Slope | 0.99 |
| Aspect | 0.97 |
| Elevation | 0.97 |
| PDSI | 1.50 |
| Landcover type | 0.0 |
| Forecast: Total precipitation | 0.0 |
| Forecast: Wind speed | 0.0 |
| Forecast: Wind direction | 0.0 |
| Forecast: Temperature | 0.0 |
| Forecast: Specific humidity | 0.0 |
| Active fire | 99.83 |

# 6 Experimental details

**Preprocessing** Originally, the active fire maps contain for each pixel the minute at which it has been detected as an active fire. We reduce this to the hour of detection, to prevent overfitting that might happen with too detailed detection times. Additionally, we compute a binary fire/no-fire mask as an input feature. Wind direction, forecast wind direction and aspect are measured in degrees in $[0, 360)°$. To those, we apply the $\sin$ function, to ensure that their extreme values are close in feature space. Since the land cover class is a categorical variable, we represent it with a one-hot encoding. We follow the approach taken by Huot et al. to standardize features to mean 0 and std 1, except for the degree features mentioned above, the land cover class, and the binary fire map. The statistics used for standardization are precomputed for each combination of years in the training set, excluding any missing values. For the fire masks, the mean reflects the mean time of detection. Pixels without active fire detections are ignored for computing mean and standard deviation. Missing values in all non-fire features are replaced with zero after standardization.

**Augmentations** During training and validation, we randomly crop the differently-sized images to $128 \times 128$ pixels, to be able to batch images of different original resolutions. We always use a batch size of 64 for training and validation, but a batch size of 1 for testing, to use the full image. As part of the cropping, we oversample fire pixels, to counteract the dataset imbalance. For that, we consider ten different random crops and keep the one with the most active fire pixels in the label. As a secondary criterion, we consider the number of active fire pixels in the input fire maps. We combine both of these criteria by simply multiplying the number of fire pixels in the label with 1000 and adding the number of fire pixels in the input fire maps. This way, we can compare different random crops based on a single scalar. Apart from cropping, we also perform horizontal and vertical flips, as well as rotations by multiples of 90° (and adjusting the degree-based features accordingly), again following Huot et al. During testing, we only center-crop the image to side lengths that are multiples of 32, which is a requirement of the U-Net implementation we use.

For UTAE, we use the day of the year as a separate input feature, which is used to compute temporal embeddings.

**Multi-temporal data for U-Net** Since default U-Net architectures can not deal with the additional temporal dimension, we concatenate the different frames along the channel dimension. We then remove the features that do not vary over time for all but the last time step. These are slope, aspect, elevation and land cover class. Since the land cover class is represented as a one-hot encoding, this greatly decreases the dimensionality of the input.

**Hyperparameter search** To determine the learning rate and loss function, we ran coarse grid searches for each model separately, using the train/val/test split of $\{2018,2020\}/2019/2021$. We combined learning rates in $\{10^{-i}|1 \leq i \leq 5\}$ with loss functions in $\{$weighted binary cross-entropy, focal loss, Dice loss, Jaccard loss$\}$. The cross-entropy and focal loss use a weight of 236 for the fire class.

**ConvLSTM testing** Since the ConvLSTM is limited to the image dimensions that it is trained on, we iteratively let the model predict adjacent crops of size $128 \times 128$. We fuse the predictions and then evaluate the result against the ground truth. Crops in the last row or column often will not fill out the full $128 \times 128$. We tried to solve this by using padding, but received results that were far from what could be expected based on the validation performance. Instead, we now align crops in the last row or column with the bottom or right edge of the image, leading to an overlap with the previous crop in that row or column. Where crops overlap, we simply overwrite the earlier predictions with the newer ones.

## 6.1 Additional ablation: Combining features

In Table 4, we show the ablation study of how cumulatively adding features to the U-Net input influences performance. The order of the features follows the performance in the single-feature evaluation ablation study, starting with the best performance and moving towards the lowest performance. Ideally, we would investigate all possible feature combinations, but apart from the high number of combinations, the high number of runs required for the 12-fold cross-validation makes this computationally prohibitive. This experiment is thus a compromise, to still estimate which feature combinations might be most useful. The performance does not monotonously rise when adding more features. The best performance is achieved when using vegetation, land cover, topography and weather features. Adding ERC, drought index, and weather forecast features does not improve the results. When including all of these features in five input days, the performance even decreases by about three percentage points. We use these results in the main part of the paper, by applying the best-performing feature combination from this ablation in the training of the non-U-Net architectures.

The results can be taken as a sign of the difficulty of optimizing in high-dimensional feature spaces. Theoretically, if the model receives additional features that are not informative, it could just learn to ignore these, setting their weights to zero. However, we see in multiple instances that the additional features lead to worse performance. The same happens more inputs are added in the temporal dimension.

Table 4: **Ablation: Combining features** Starting from NDVI, EVI2 and VIIRS observations (vegetation) and the fire masks, we cumulatively add more features, going from the left side of the table to the right. The rightmost column represents an experiment with all available features. The models predict the wildfire spread 24h ahead, based on one to five days of observations. The performance displayed is the **mean test set average precision ± the standard deviation**, computed in a 12-fold cross-validation over years. Higher is better.

| Input days | $\rightarrow$ Cumulative features $\rightarrow$ | | | | | |
|---|---|---|---|---|---|---|
| | Vegetation | Topography | Landcover | Weather | ERC, drought | Weather forecast |
| 1 | $0.316 \pm 0.091$ | $0.321 \pm 0.081$ | $0.315 \pm 0.086$ | $\mathbf{0.329 \pm 0.086}$ | $0.327 \pm 0.091$ | $0.328 \pm 0.090$ |
| 5 | $0.323 \pm 0.081$ | $0.316 \pm 0.081$ | $0.322 \pm 0.084$ | $\mathbf{0.334 \pm 0.076}$ | $0.333 \pm 0.090$ | $0.303 \pm 0.111$ |

## 6.2 Strong variance in the results

All results show unusually high standard deviations of about 8-11 percentage points, while the mean is around 30 percentage points of average precision (AP). The reason for this is that the different

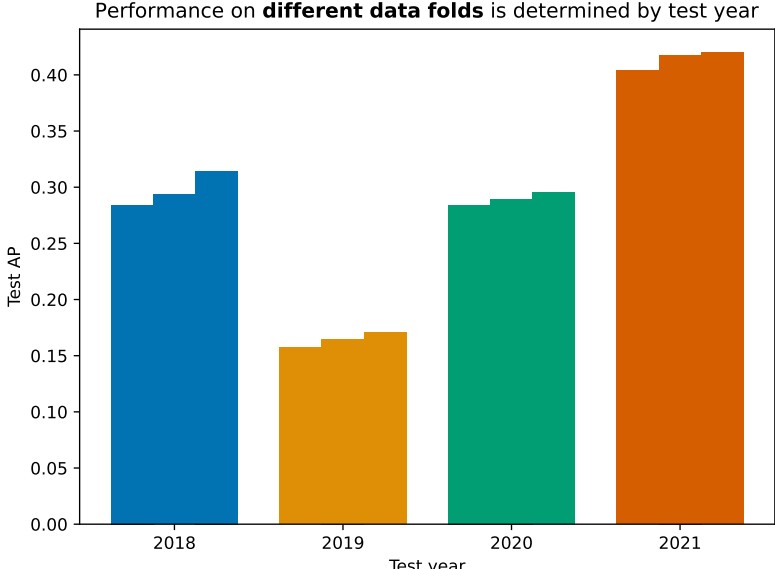

Figure 2: **Performance varies strongly between data folds** When training the U-Net only on fire masks and the VIIRS reflectance features, the per-fold test performances cluster based on the test set. Each bar represents test results on one data fold. The grouped bars represent folds that have the same test year, but vary in their assignments of training and validation years. Within each group of three bars, we sort the bars in order of ascending performance, for easier visual inspection. The models achieve consistently low performance on the test year 2019 and high performance on 2021. The models predict the wildfire spread 24h ahead, based on one days of observations. Higher is better.

Table 5: **Data folds used** For clarity, we present all data folds used. They cover all permutations of the four years into a set of two training years and two sets of one validation and one test year.

| id | Train years | Val years | Test years |
|----|-------------|-----------|------------|
| 0 | (2018, 2019) | 2020 | 2021 |
| 1 | (2018, 2019) | 2021 | 2020 |
| 2 | (2018, 2020) | 2019 | 2021 |
| 3 | (2018, 2020) | 2021 | 2019 |
| 4 | (2018, 2021) | 2019 | 2020 |
| 5 | (2018, 2021) | 2020 | 2019 |
| 6 | (2019, 2020) | 2018 | 2021 |
| 7 | (2019, 2020) | 2021 | 2018 |
| 8 | (2019, 2021) | 2018 | 2020 |
| 9 | (2019, 2021) | 2020 | 2018 |
| 10 | (2020, 2021) | 2018 | 2019 |
| 11 | (2020, 2021) | 2019 | 2018 |

years in the dataset seem to have very different distributions. Figure 2 shows the full results for training a U-Net on the fire masks and VIIRS reflectance features, separated into the individual fold performance, that we usually aggregate over. All folds that test on the year 2019 receive much lower performance than average, while testing on the year 2021 leads to much higher performance than average. We interpret this as an indication that the data distribution varies strongly across years. However, this does not seem to have a strong impact on training, seeing how the bars that are grouped together all have rather similar performance. Part of this is likely explained by the fact that the training set is always composed of two years, thereby dampening adverse effects that a single unusual year might have. Table 5 shows the different folds we used.

## 6.3 Feature importance: Linear regression

As a very basic indicator of feature importance, we present the aggregated weights of the linear regression models in Table 6. Most features are standardized in pre-processing. Some are not, which must be considered when comparing the magnitude of weights between features. Namely, land cover classes are one-hot encoded; the binary fire mask consists only of binary values; and wind direction, forecast wind direction and aspect take values in [-1,1].

We see that precipitation and certain land cover classes (croplands, urban areas, shrublands, water bodies, snow) are features that indicate no fire at the center of the $3 \times 3$ pixel window on the next day. The strongest indicators of active fire on the next day are the active fire features of the current day, which is expected. If no fire is present, there is only a small chance for a new fire to spawn the next day. Additionally, savannas and evergreen forests contribute to the model predicting active fire. The presence of shrublands as a negative indicator is surprising, since they should be prone to fire. We assume that those fires might be too small to show up as consistently-spreading wildfires in our dataset, leading to negative weights in the linear regression models.

Interestingly, land cover classes have a much higher influence on the linear progression model than other fuel features like NDVI, which is an indicator of living vegetation, i.e. a fuel indicator with a much higher temporal resolution than the land cover class. The reason is likely that classes like urban or evergreen forest clearly indicate a very low or very high amount of fuel based on their definition, even if they do not have a high temporal resolution.

Other features, like the drought index PDSI, the VIIRS bands, or the topographical features are largely ignored by the model. This disregard could indicate that they are not useful, provide redundant information, or that they are not immediately usable by the very simple linear regression model. For example, topography data should become more useful when combined with the wind direction or multi-temporal data about the direction in which the fire spreads.

## 6.4 Runtimes

In Table 7, we compare the runtimes per batch of different models during training and testing. During testing, all models use a batch size of 1, since the image resolution changes between different fires. The runtimes were computed on a shared cluster, using one A40 GPU each, and are thus impacted by external factors that are outside of our control, like reduced disk I/O due to other users' processes.

Table 6: **Feature importance:** We compute the feature importance as the mean weight associated with each feature in the linear regression model. The model has a $3 \times 3$ pixel input window, thus nine weights associated with each feature. We compute the mean of those nine values. Then, we compute the mean and standard deviation of those features across all twelve data folds. Positive values contribute to predicting the center pixel as burning the next day, negative values contribute to predicting no fire.

| Feature | Mean | Standard deviation |
|---|---|---|
| Land cover: Croplands | -28.961 | 23.247 |
| Land cover: Urban and Built-up Lands | -25.943 | 11.192 |
| Total precipitation | -22.014 | 7.879 |
| Land cover: Open Shrublands | -21.324 | 23.798 |
| Land cover: Cropland/Natural Vegetation Mosaics | -14.093 | 10.664 |
| Land cover: Water Bodies | -10.620 | 9.549 |
| Forecast: Total precipitation | -9.865 | 3.767 |
| Land cover: Permanent Snow and Ice | -8.004 | 1.405 |
| Land cover: Closed Shrublands | -4.173 | 9.030 |
| Land cover: Barren | -2.043 | 6.919 |
| Land cover: Permanent Wetlands | -1.731 | 4.609 |
| Forecast: Specific humidity | -1.532 | 1.162 |
| Land cover: Grasslands | -1.532 | 1.582 |
| VIIRS band I2 | -1.347 | 1.755 |
| Land cover: Mixed Forests | -1.143 | 10.697 |
| Wind direction | -0.633 | 0.776 |
| Palmer drought severity index (PDSI) | -0.581 | 1.034 |
| VIIRS band I1 | -0.514 | 1.350 |
| Land cover: Deciduous Needleleaf Forests | -0.367 | 0.949 |
| EVI2 | 0.023 | 1.450 |
| Wind speed | 0.121 | 1.100 |
| Forecast: Wind direction | 0.233 | 0.841 |
| Forecast: Wind speed | 0.260 | 1.522 |
| Aspect | 0.457 | 1.384 |
| Land cover: Deciduous Broadleaf Forests | 0.575 | 6.292 |
| Minimum temperature | 0.690 | 1.882 |
| Specific humidity | 0.857 | 1.030 |
| Maximum temperature | 0.948 | 1.027 |
| Forecast: Temperature | 1.519 | 0.906 |
| VIIRS band M11 | 2.156 | 1.292 |
| Slope | 2.406 | 0.839 |
| Energy release component | 2.637 | 2.128 |
| Elevation | 2.933 | 2.117 |
| NDVI | 4.178 | 3.244 |
| Land cover: Savannas | 5.165 | 2.775 |
| Land cover: Woody Savannas | 6.602 | 2.082 |
| Land cover: Evergreen Needleleaf Forests | 7.344 | 2.152 |
| Land cover: Evergreen Broadleaf Forests | 9.106 | 6.972 |
| Active fire | 16.531 | 3.817 |
| Active fire (binary) | 24.217 | 7.783 |

Table 7: **Model runtimes per batch:** We compare the runtimes of all models on data fold 0, with the test set adjusted to be comparable with a window of five leading observations. For the test set, this includes only the prediction, with a batch size of 1, since different fires have different image proportions. For the train set, this includes a full training epoch with backward passes and weight updates. The ConvLSTM test prediction times are higher than those of other models, because this model requires several predictions for each input, due to the model being limited to a fixed input image resolution that differs from the test time resolution.

| Model | Leading obs. | Test batch size | Test pred. time | Train batch size | Train time |
|---|---|---|---|---|---|
| Log. Regression | 1 | 1 | 0.029s | 64 | 0.167s |
| Res18 U-Net | 5 | 1 | 0.035s | 64 | 0.645s |
| ConvLSTM | 5 | 1 | 0.185s | 32 | 0.395s |
| UTAE | 5 | 1 | 0.091s | 32 | 0.435s |