# OpenReview forum: "WildfireSpreadTS: A dataset of multi-modal time series for wildfire spread prediction"
_NeurIPS.cc/2023/Track/Datasets_and_Benchmarks — NeurIPS 2023 Datasets and Benchmarks Poster_

### Official Review · Reviewer_3tFe · 2023-07-19
**Multi-sensor multi-temporal dataset for wildfire spread forecasting**

**Rating:** 6
**Confidence:** 4

**Strengths:**

### Significance:
* Wildfires are devastating disasters of increasing frequency, whose occurrence should be mitigated. This submission proposes a dataset for forecasting wildfire spread with time series of up to five days, whereas most prior work thus far focused on live detection or single day time horizons. I consider the extension of the temporal component is a meaningful improvement over prior work.
* The submission showcases its proposed dataset by benchmarking five fundamentally different models, plus a total of 10 variations thereof: two simple approaches, three convolutional models with a U-Net backbone, three LSTM networks and two attention-based models based on U-TAE. The different variants are designed to investigate the benefits of multi-temporal data, plus the opportunities and challenges of utilizing high-dimensional multi-sensor information for wildfire spread forecasting.

### Relevance:
* The occurrence of wildfires increased over the past few years and is prognosed to rise further. Topicwise, this makes wildfire monitoring an important matter. Accordingly, the subject has grown of scientific relevance recently, e.g. with a handful of accepted papers on the topic at last year’s [NeurIPS Workshop on Tackling Climate Change with Machine Learning](https://nips.cc/virtual/2022/workshop/49964). My impression is that a growing extent of the core machine learning community becomes interested in AI for social good and, per extension, utilizing satellite data for such purposes.

### Accessibility and Accountability:
* The proposed WildfireSpreadTS dataset will be released on Zenodo, which is a common and accessible choice for open data sharing in the scientific community. The dataset is only released officially upon acceptance, but the authors share a private version with the reviewers, plus the provided Datasheet for Datasets gives a good overview of the curated dataset.
* Moreover, I appreciate the authors share both their code for [using the dataset and baselines](https://github.com/SebastianGer/WildfireSpreadTS) as well as for [re-creating the WildfireSpreadTS dataset](https://github.com/SebastianGer/WildfireSpreadTSCreateDataset). The latter may also be beneficial for the community to modify and extend the proposed data, e.g. to different geospatial territory.

### Ethical and social implications:
* Research on a timely forecasting of wildfire spread may in the future yield better predictions and improved early warning systems, whose social implications would be clearly positive.

**Additional Feedback:**

* A further analysis of the dataset’s class statistics would be informative, to better understand potential bias in the data and the challenges of the considered task. Specifically, what is the percentage of fire, non-fire and missing data pixels in the dataset? What is the distribution of pixelwise delta, i.e. how much change (particularly from non-fire to fire-pixels) occurs on a frame-by-frame basis?
* The submission makes it very clear that, in addition to temporal resampling to a regular 24h spacing, “All features were resampled to the spatial resolution of the active fire maps of 375m” [p.3, Fig.1 subtext]. Similarly, the prior work of Huot et al. [2022] resampled their multi-sensor data to 1km, the spatial resolution of their labels. Have you ever considered to upsample each input sensor’s data to the best modality’s resolution instead (and eventually increase the spatial context), in order to not lose any information in the spatial domain? I am wondering whether this may preserve any information that the deep networks might extract for better spread forecasts, even when predictions are finally evaluated in a more coarse resolution. Likewise, it might have been interesting to further downsample the input data to the 1km resolution previously utilized by Huot et al. [2022], in order to get an estimate of how much more helpful the improved spatial resolution of 375m may be.
* Please also report the parameter count and runtime complexities for the considered baselines, e.g. in the supplementary material. This may facilitate contextualizing the benchmarked methods, in particular when involving attention-based backbones.

**Clarity:**

The paper is well written. As mentioned previously, the authors are very explicit with outlining encountered problems and limitations to the best of their knowledge, which is appreciated and should be positively taken note of.

**Correctness:**

* My main concern regarding the submission is with the mixed, and at times poor performances of baselines evaluated on the WildfireSpreadTS dataset and the proposed forecasting task. This issue also relates to the optimization difficulties acknowledged in section 5.4 of the main text, and my concerns previously stated in section “Weaknesses: Significance”. In brief, I find it hard to disentangle the lack of suitable models and their mixed results from potential issues with the dataset’s design or task. A key outcome I find particularly troublesome, is that the only temporal baseline beating the trivial persistence approach in terms of test F1 score is the Res18 U-Net (but solely when using parts of the proposed feature modalities). This makes me doubt the choice of baselines (see: “Weaknesses: Significance”), meaningfulness of input features (the curse of dimensionality seems to outweigh their usefulness) or proper task design (would there have been fewer issues encountered when choosing sequences of lengths 2,3,4, … instead?). Clearly, it is not the authors’ obligation to design a new approach that performs SOTA on their suggested task, but a trend of well-behaving optimization plus decent performances would at least confirm the dataset’s and the proposed task’s good design, which I can’t take for granted yet based on the experimental results. Following are more specific questions:
    * As the selection of considered baseline models intersects to some extent with that of the prior work by Tarasiou et al. [2023], have similar issues been reported previously?
    * In section “5.4 Optimization difficulties” it is explained that the high-dimensional input space in conjunction with time series data poses a severe challenge to optimizing e.g. the considered UTAE or TSViT baselines. I am skeptical whether this truly is a sufficient explanation specifically for UTAE, whose U-Net backbone performs a parallel encoding of each time point individually. Thereby, the effective dimension of input features to be processed by the first encoder layer does not scale with the time series length. That said, both evaluated UTAE setups nonetheless perform worse than the persistence baseline in terms of F1 score, and the variant utilizing all the curated modalities suffers from the reported optimization instabilities.
* The reported results show considerable performance gaps and fluctuations between train, validation and test splits, which are defined in a cross-annual manner. This is acknowledged by the authors, and it is suggested that “users think about other setups, for example a cross-validation approach.” [supmat, p.2, l.63-64]. I second that e.g. leave-one-out cross-validation would have been a more appropriate choice to abstract away from the observed annual fluctuations. However, I clearly consider doing so (and thereby proving that outcomes on the designed dataset are not overly affected by arbitrary split decisions) is in the authors’ responsibility (as in Garnot and Landrieu [2022]), rather than deferring this tedious task to the future userbase of the dataset.

In sum, I appreciate the submission’s motivation and core idea, which certainly has potential. However, the persistent challenges of optimizing a representative variety of baselines on the proposed dataset in combination with the mixed experimental results and the uncertainty of what’s all involved in causing these issues poses an obstacle for me to recommend the submission for acceptance in its current version. The outlined concerns will hopefully serve the authors well in addressing any issues.

**Documentation:**

The authors do a good job properly documenting their dataset, e.g. in the framework of the Datasheet for datasets. Furthermore, not only do the authors provide code for [using the dataset and reproducing baseline performances](https://github.com/SebastianGer/WildfireSpreadTS), but also for [re-creating the WildfireSpreadTS dataset](https://github.com/SebastianGer/WildfireSpreadTSCreateDataset).

Based on this, I consider the submission is well documented. Furthermore, I suppose sufficient detail is provided to re-evaluate the considered baseline models as proposed by the authors. However, see my concern w.r.t reproducibility in section “Weaknesses: Accessibility and Accountability” regarding discrepancies in data and section 5.4 of the main text pertaining to optimization difficulties (partly, some of the models have a very large spread across repeated runs).

**Ethics:**

There are no direct concerns.

**Limitations:**

I appreciate the authors’ practice of making any encountered problems or limitations very explicit, to the best of their knowledge. It is unfortunate that I do in fact find some of the outlined issues problematic, but having them fairly exclaimed before any reviewer first brings them up hopefully makes the review process more agreeable for all involved parties, and maybe the responsible chairs want to take this into account in a suitable manner. In particular, here's two limitations mentioned before, which I'd like to highlight in this text box:

> [...] the reported results are overly mixed and make me wonder about the suitability of the chosen baselines---or alternatively, the appropriateness of the curated data and the proposed task. Generally encountered optimization problems are acknowledged in section 5.4 of the main text, while the performances of some baselines may also be affected due to model-specific experimental design choices („likely influenced by how we have to compute test set predictions for this specific architecture, using crops of padded images to follow the image size requirements prescribed by the architecture.“ (p.8, l.272-274)). Overall, this seems to me like part of the baselines may have been chosen or handled unfortunately, but it is challenging to rule out further issues.

This is the most apparent issues I'm having with assessing this submission. As stated by the authors in section 5.4 of the submission, the proposed dataset poses several challenges (heterogeneous multi sensor data, high dimensional input space, time dimension on top) that make it difficult to optimize established baselines upon. One potential way of interpreting this issue is to conclude that a broad range of existing methods are not suitable for the submitted dataset and hence further methodological development is needed. However, I think this would be oversimplifying the problem and that there's still need for the proposed dataset to prove itself: notably, parts of its significance is due to more data modalities (which may cause the high-dimensional issues and result in poorer optimization than using fewer modalities) and the time series extension (which oftentimes causes optimization problems and yield mixed results). The drawbacks encountered at the cost of introducing these novelties make it challenging to fully appreciate them. It is challenging to propose any solution to this issue, but I think one initial step would be to run ablations determining the benefits of the new modalities individually, plus considering shorter time series of lengths 2, 3, 4 and analyzing at which point the benefits of additional data are offset by the challenges of optimization.

>It is my understanding that (parts of) the benchmarks are based on a digital elevation model that may not be accessible in the dataset release, which raises concerns about the reported outcomes being faithfully reproducible. Concretely, the main text states „Due to a misunderstanding, the experiments in this paper use the SRTM90_V4 product Reuter et al. [2007] by CIAT as the source of topographical data. The published dataset contains the NASA SRTM V3 product instead“ [p.3, l.145-147]. I appreciate the authors’ transparency and see no strong reason to doubt their estimate this wouldn’t significantly change the reported outcomes. However, the discrepancy of data being used in the paper versus data being released in the dataset makes me wonder whether the reported results would still be reproducible, which may be considered an issue.

This is an issue which should ideally not occur in a submission to a selective venue, but I'd not consider it a reason for rejection on its own. As suggested above: Alternative to re-computing the experiments based on the correct elevation maps, would it be feasible to also release the utilized SRTM90_V4 with the final dataset? I could imagine that future users of the dataset would rather prefer to faithfully stick to the data as utilized in the experiments, so they may be more confidently comparing their results with the reported outcomes. I think that addressing this issue, at least to some extent, shouldn't be considered totally impractical.

**Opportunities For Improvement:**

### Significance:
* The submission’s main contribution, WildfireSpreadTS, is foremost a collection of already available data publicly accessible via Google Earth Engine. However, curating and selecting a subset of observations from publicly accessible data of multiple sources is a common practice for constructing novel remote sensing datasets.
* To showcase the dataset, a set of five fundamentally different models are evaluated. Combined with the considered network variations, I think this is a fair count of baselines for a submission whose primary focus is not on benchmarking. However, the reported results are overly mixed and make me wonder about the suitability of the chosen baselines---or alternatively, the appropriateness of the curated data and the proposed task. Generally encountered optimization problems are acknowledged in section 5.4 of the main text, while the performances of some baselines may also be affected due to model-specific experimental design choices ( „likely influenced by how we have to compute test set predictions for this specific architecture, using crops of padded images to follow the image size requirements prescribed by the architecture.“ (p.8, l.272-274)). Overall, this seems to me like part of the baselines may have been chosen or handled unfortunately, but it is challenging to rule out further issues. Please also see my concerns stated in section “Correctness”, related to this issue.

* Part of WildfireSpreadTS’s appeal is that it provides additional information that was not contained in prior datasets. This includes weather forecast data, digital elevation model features and land cover types. The authors do a good job of theoretically motivating their inclusion, but my complaint is that the practical benefits of including such information are not yet demonstrated. It would have been desirable to see e.g. via ablation studies how each of the newly curated features benefits a better wildfire spread prediction, as previously done in Tables IV-V "Feature Ablation" of Huot et al (2022). As it currently stands, the reader may not yet sufficiently appreciate the importance and contribution of individual input modalities, including the ones newly curated by the authors to differentiate their work from preceding datasets --- which makes this innovation’s significance rather speculative. Seeing the newly included modalities prove beneficial is particularly critical in light of the issue raised above, with additional input feature dimensions rendering the optimization process challenging or unstable.

### Relevance:
* The data collected in the proposed dataset are constrained to wildfires from years 2018-2021 in the United States. The geospatial focus excludes other parts of the world and begs to ask whether the reported outcomes or networks trained on WildfireSpreadTS may generalize to different countries with fuel properties, fire dynamics and climate distinct from the United States. Moreover, the reported outcomes showed considerable performance discrepancies across the four included years and further fluctuations may exist beyond the considered time span.

###  Accessibility and Accountability:
* It is my understanding that (parts of) the benchmarks are based on a digital elevation model that may not be accessible in the dataset release, which raises concerns about the reported outcomes being faithfully reproducible. Concretely, the main text states „Due to a misunderstanding, the experiments in this paper use the SRTM90_V4 product Reuter et al. [2007] by CIAT as the source of topographical data. The published dataset contains the NASA SRTM V3 product instead“ [p.3, l.145-147]. I appreciate the authors’ transparency and see no strong reason to doubt their estimate this wouldn’t significantly change the reported outcomes. However, the discrepancy of data being used in the paper versus data being released in the dataset makes me wonder whether the reported results would still be reproducible, which may be considered an issue. It might be a matter of licensing, but the obvious solution to ensure replicability here would be to just as well release the utilized SRTM90_V4 with the final dataset. Would this not be feasible?

### Ethical and social implications:
* I don’t see any immediate negative ethical or social drawback.

**Relation To Prior Work:**

Relation to prior work is discussed in sections 3 and 3.1 of the submission. The closest preceding publications are by Huot et al. [2022] and Jeffrey et al. [2023], proposing an aggregated or one day lead time wildfire prediction task, extended by VIIRS active fire maps. The submission does a good job of outlining the differences of WildfireSpreadTS to these preceding publications. Primarily, this includes time series data (multi-day inputs and more frequent observations) as well as additional features (such as weather forecasts, digital elevation properties and land cover types). Intuitively, the supplementary information sounds beneficial over what has previously been provided, e.g. by the NextDayWildfireSpread dataset of Huot et al. [2022].

**Summary And Contributions:**

**Update:** Following the rebuttal process, I decided to **raise the rating by one, to a score of 6**. This is thanks to the submission having undergone substantial improvements, including
- re-evaluation of all considered baselines in a cross-validation experimental setup. By averaging out annual fluctuations in the dataset the reported results became much more clear and in line with what one may intuitively expect. The originally very noisy outcomes were the cause of my main concerns, raising questions about whether the ambiguous results would be due to inappropriate baselines or a potentially poor design of the dataset. This issue has been addressed and dealt with.
- experiments are re-run with the correct digital elevation model data. The originally submission mixed up two comparable products in a manner which made me question the work’s reproducibility. This problem has been fully resolved.
- finally, ablation studies on varying input time lengths and subsets of input features are conducted to underline the respective data’s importance. The submitted dataset’s key contributions are its multi-sensor and multi-temporal features, yet my complaint was that their relevance has not been sufficiently demonstrated originally. This has been addressed by a series of new experiments.

In sum, I consider it appropriate raising the rating and recommending the submission for publication. Thanks to the authors for a constructive discussion.

---

The submission provides **WildfireSpreadTS, a new dataset for wildfire spread prediction**. The dataset consists of time series of multi-sensor satellite data---including VIIRS optical imagery, vegetation indices and weather as well as downstream products such as fire masks and land cover maps, all harmonized to a common spatio-temporal framework. Key contributions are highlighted in **bold** font.

The work is well motivated, and I consider the dataset’s temporal forecasting focus a meaningful extension of prior work. Moreover, applying machine learning for disaster management is an important topic that becomes of growing relevance. However, the submission suffers from a few limitations and an experimental section of mixed results, which make it challenging to assess the choice of benchmarked baselines and the benefits of the curated data. As specified in this review’s sections “Weaknesses: Significance” and “Correctness”, ambiguous baseline performances make me question the perks of the collected multi-sensor data, the optimization over time series or the design of the proposed forecasting task. In sum, I have mixed feelings about the paper’s experimental parts and hence would not feel confident recommending this submission for publication in its present form with the outlined concerns applying, even though the work has potential. Unfortunately, attributing the unclear experimental outcomes solely to a lack of suitable baselines is as challenging as ruling out potential issues with the design of the dataset or the proposed task, so I choose to lower my rating’s confidence to “fairly confident”, albeit being quite familiar with remote sensing research.

---

> ### Author Response · Authors · 2023-08-18
>
> > “I think one initial step would be to run ablations determining the benefits of the new modalities individually, plus considering shorter time series of lengths 2, 3, 4 and analyzing at which point the benefits of additional data are offset by the challenges of optimization. [...]
> e.g. leave-one-out cross-validation [...]”
>
> Based on the feedback by several reviewers, we have basically done exactly this. See the note to all reviewers for details. The results are now much more reliable, due to the cross-validation, and the effect that different features have much more clearly discernible, due to the ablation studies. This feedback (and similar one by other reviewers) has greatly improved the quality of the paper.
>
> > SRTMV3 vs V4
>
> We re-ran all of our experiments, so this is not an issue anymore.
>
> > “Specifically, what is the percentage of fire, non-fire and missing data pixels in the dataset? What is the distribution of pixelwise delta, i.e. how much change (particularly from non-fire to fire-pixels) occurs on a frame-by-frame basis?”
>
> The percentage of fire and missing data per feature will be added to the supplementary materials within the next week. For fire pixels, we could represent the change as percentages: “How likely is it that a fire becomes a non-fire in the next frame?” and the other way around. Would that address your comment properly?
>
> > “Have you ever considered to upsample each input sensor’s data to the best modality’s resolution instead (and eventually increase the spatial context), in order to not lose any information in the spatial domain?”
>
> We have initially thought about different ways to deal with the strongly varying resolutions in the data sources. The conclusion was that the label data takes precedence over all other data and thus everything has to follow the label’s resolution. Alternatively, we could get 10m resolution for some of the input data (land cover, topography), meaning that we have one binary classification target for an area of 38x38 pixels. Whether this high resolution in very few of the input features would help to improve the overall binary prediction for each patch of 38x38 is something that would have to be investigated separately. To us, the resampling to the resolution of the label seemed the most sensible compromise for now, also considering the impact that this would have on the size of the dataset. If more features were available at higher resolution, this might become more feasible. Alternatively, approaches like in WildfireDB could be considered, summarizing the higher resolution data before fusing it with the lower resolution data. Self-supervised pretraining to create low-dimensional embeddings of the high-resolution data might work even better than the human-defined statistics used in WildfireDB. In both cases, we would have a large increase in the number of features, which is already a problem with the dataset in its current form.
>
> > “Likewise, it might have been interesting to further downsample the input data to the 1km resolution previously utilized by Huot et al. [2022], in order to get an estimate of how much more helpful the improved spatial resolution of 375m may be.”
>
> The direct comparison with a MODIS product by simply downsampling the image resolution might be less meaningful than initially expected, since the MODIS product seems to have its own issues with false positive detections. This is based both on our observation that MODIS-based GlobFire data often does not coincide with the higher resolution VIIRS active fire data, as well as [Masocha et al.: Accuracy assessment of MODIS active fire products in southern African savannah woodlands](https://onlinelibrary.wiley.com/doi/full/10.1111/aje.12494). This potential label noise in the MODIS product could have influenced Huot et al.’s result, but would not influence our downsampled data, leading to overly optimistic results on our side. We did, however, use Huot et al.’s best one-feature setting of NDVI + fire mask from their Table V as a starting point to determine how much benefit different features provided.
>
> > “Please also report the parameter count and runtime complexities for the considered baselines, e.g. in the supplementary material. This may facilitate contextualizing the benchmarked methods, in particular when involving attention-based backbones.”
>
> The parameter count had been reported in the supplementary materials in Table 2. We have now moved it into Table 5 in the main paper, to better contextualize the different models’ performances. Could you elaborate on what kind of runtime complexity you are interested in and why, so we can address it adequately? Since we are working at a temporal resolution of 24h, inference time did not seem like a critical feature to us so far. We could, for example, give the wall clock time that a model needs to perform inference on a validation set of X images with batch size Y, using a specified GPU and number of CPU cores.

---

> > ### Comment · Reviewer_3tFe · 2023-08-27
> > **Reviewer reply**
> >
> > Dear authors, I appreciate the strong rebuttal + revision and have increased the score by one. Concerning your questions:
> >
> > > For fire pixels, we could represent the change as percentages: “How likely is it that a fire becomes a non-fire in the next frame?” and the other way around. Would that address your comment properly?
> >
> > Yes, how it is now included into the supplementary material is a suitable format. Thanks for the additional information.
> >
> > > Could you elaborate on what kind of runtime complexity you are interested in and why, so we can address it adequately? Since we are working at a temporal resolution of 24h, inference time did not seem like a critical feature to us so far.
> >
> > It is of no great importance for the rebuttal, but in the future the target audience may implement models based on e.g. ViT or spatio-temporal backbones such as [i]. To facilitate the comparison of further models with the considered baselines, having an overview of computation complexities as in eq (11) of [i] available may be helpful. It is not necessary, but you may want to consider including such information into the supplementary material for the camera-ready version.
> >
> > ---
> >
> > *[i] Zeng, Y., Fu, J., & Chao, H. (2020). Learning joint spatial-temporal transformations for video inpainting. In Computer Vision–ECCV 2020: 16th European Conference, Glasgow, UK, August 23–28, 2020, Proceedings, Part XVI 16 (pp. 528-543). Springer International Publishing.*

---

### Official Review · Reviewer_KxL4 · 2023-07-20
**This is a well written paper, centered around active fire prediction, conditioned on a set of multimodal wildfire drivers.  In my opinion the paper makes a contribution to the community, but it is marginal with respect to what is already available. There are two major shortcomings, 1) the machine-learning datasets used in the experimental section are not provided,  2) Added-value with respect to previous work (WildfireDB) is not mentioned at al.)**

**Rating:** 7
**Confidence:** 5

**Strengths:**

The paper is focused on a very important and relevant topic in view of climate change. It provides an harmonised dataset of wildfire drivers
to perform active prediction, using multi-temporal wildfire drivers as input. Applications based on this dataset have the potential to provide actionable knowledge to emergency responders and civil protection authorities. In addition, it can provide valuable insights to the scientific community on wildfire behavior and spread patterns. The dataset has also the potential to study the evolution of extreme wildfire events.
A nice feature of the dataset is the inclusion of weather forecasts, not just reanalysis meteorological data. In this way distribution shifts between training and production can be avoided.

**Additional Feedback:**

I think that some additional work is needed to make the dataset ML-ready to allow benchmarking. Currently there are many different ways the dataset can be exploited, which is fine, but developing a couple of ML tracks would make it more focused and would foster the uptake of the dataset by the scientific community.

**Clarity:**

The paper is well written. I would propose that the experimental section is shortened and a ML dataset extraction subsection is added. Currently the dataset as I understand it is not ML ready, e.g. input images are of variable input dimensions. The paper should include information on how the ML samples are extracted from the WildfireSpreadTS dataset (perhaps move some text from the supplementary material in the main text).
A figure of the ML datasets (multi-temporal and unitemporal), plus the target variable evolution would be helpful as well.
Figure 3 is not clear, please revise.
Some references styling needs fixes, e.g. lines 17-18.
For burnt areas it is better to use hectares instead of square meters.
Lines 98-102 need some elaboration.

**Correctness:**

The claims are correct. The dataset is constructed in a sound way with respect to the granularity of the variables - all are harmonized to the same spatio-temporal resolution. Image dimension however vary, which may be a shortcoming in the use of the dataset, since researchers may adopt different approached to crop/sample the dataset. This makes benchmarking more difficult.
I would suggest to include AUROC plots for the evaluation as well, since this provides better insights in the overall model performance.

**Documentation:**

Limited information about the data sources are included in the github repo. The paper includes the details for the data collection.

**Ethics:**

No ethical concerns.

**Limitations:**

The authors are clear in the limitations of their work, especially for the experimental section.

**Opportunities For Improvement:**

1) The datasets misses out on an opportunity to provide a more holistic contribution to wildfire forecasting, by including fire ignitions from VIIRS (as already done), fire radiative power and burnt areas (these were already used to filter the VIIRS hotspot data).
2) It would be nice to include proxies for anthropogenic drivers (e.g. road network, populated areas, fire brigade locations, etc.) for wildfire ignition and spread, next to climatic, vegetation and landscape.
3) The dataset is not ML-ready (e.g. variable image sizes), special care should be taken to create training/validation/testing samples.
4) Some information/insights about the noise in the labels would be appreciated.
5) It would be nice to have experiments without reflectance data, but include only the spectral indices. Perhaps the latter are a good enough proxy for the problem (e.g. see Kondylatos et al.) and this reduces computational complexity?

**Relation To Prior Work:**

There is a clear discussion on how this work differs from the "Next Day Wildfire Spread” paper, although in my opinion this is marginal. The key added value is the inclusion of meteorological forecasts.
However, the authors do not mention the previous NeurIPS dataset WildfireDB (https://openreview.net/forum?id=6nblryHxVbO), that also uses VIIRS hotspot data and similar input drivers.

**Summary And Contributions:**

The dataset is centered around active fire prediction, conditioned on a set of multimodal wildfire drivers. The dataset consists of 13 607 images across 607 fire events in the United States spanning over 4 years (2018-2021). The granularity of the 23 input variables is 24h temporal resolution and 375 m spatial resolution. The setup of WildfireSpreadTS allows for time-series forecasting.

---

> ### Author Response · Authors · 2023-08-18
>
> > “The datasets misses out on an opportunity to provide a more holistic contribution to wildfire forecasting, by including fire ignitions from VIIRS (as already done), fire radiative power and burnt areas (these were already used to filter the VIIRS hotspot data).”
>
> The dataset in its current form is meant as a first step, but including FRP and burnt areas are definitely good options for future work. The included VIIRS bands can already provide some distinction between burnt and unburnt areas.
>
> > “It would be nice to include proxies for anthropogenic drivers (e.g. road network, populated areas, fire brigade locations, etc.) for wildfire ignition and spread, next to climatic, vegetation and landscape.”
>
> Contrary to some related work, we purposefully omitted anthropogenic drivers.  As far as we are aware, these factors are mainly important to the ignition of fires, since human activity in various forms is often the cause of wildfires. We wanted to avoid these difficulties inherent to risk prediction and focus on predicting the spread of already existing wildfires. We do, however, provide the land cover feature, which includes “Urban and Built-up Lands” that can be seen as a binary proxy for human activity.
>
> > “The dataset is not ML-ready (e.g. variable image sizes), special care should be taken to create training/validation/testing samples. [...]  researchers may adopt different approached to crop/sample the dataset. [...] I would suggest to include AUROC plots for the evaluation as well.”
>
> Since various cropping strategies are part of modern image augmentation practices already, we want to provide users with the full data. Leaving the option for researchers to adopt different cropping strategies could allow for different approaches to handle the class imbalance, for example, and thereby open up space for innovation. The only requirement to be able to compare their results to ours is to predict the whole images on the test set, center-cropped to multiples of side length 32, based on a limitation imposed by one of our methods.
>
> In terms of training/validation/testing samples, we have now adopted the approach of 12-fold cross validation, to get a more accurate estimate of model performance.
>
> We have also considered your advice about adding tracks and believe that it is a very good approach to provide a clearer starting point and framework for readers to get started with. We have not adopted it for this paper, because the decision of which features to include for which tasks, and which tasks might be easy or difficult are not very clear to us yet. From our point of view, this dataset opens the stage for more exploration of what is possible, what is hard, what is useful for solving various tasks in the multi-temporal context. After we gained more knowledge, and maybe after including additional interesting features that can also be treated as targets, like the FRP and burned area you suggested, it makes more sense to define multiple tracks. We could, for example, imagine multi-task prediction setups that both read in and predict active fires, FRP, burned area at each time step.
>
>
> > “Some information/insights about the noise in the labels would be appreciated.”
>
> We have added information on noise in the AF product in section 5.1 in the supplementary material.
>
> > “The paper should include information on how the ML samples are extracted from the WildfireSpreadTS dataset (perhaps move some text from the supplementary material in the main text).”
>
> We have added information on how the cross-validation is performed and standardization is performed in section 5.1. The supplementary material in section 6 goes into further details regarding preprocessing, augmentations and how we choose crops during training. The multitemporal data is extracted in the same way as the mono-temporal data, except that multiple subsequent days are extracted, instead of just one. Each fire in the dataset is represented as one big tensor of shape [days, feature_channels, height, width]. What changes between mono- and multi-temporal is just how many days we index in the first dimension.
>
> > “A figure of the ML datasets (multi-temporal and unitemporal), plus the target variable evolution would be helpful as well.“
>
> Which information would you like to see visualized in the figure? We will add an example of how active fire detections look for several days in a row to the supplementary material to account for the second part of your request.
>
> > “Figure 3 is not clear, please revise”
>
> We will revise the figure within the next week. The in-image caption does seem less clear than it should be.
>
> > “Lines 98-102 need some elaboration.”
>
> Those lines were missing the context information that GlobFire is based on MODIS. We have now added it.
>
> > “However, the authors do not mention the previous NeurIPS dataset WildfireDB, that also uses VIIRS hotspot data and similar input drivers.”
>
> We have added WildfireDB in the related work section.

---

> > ### Comment · Reviewer_KxL4 · 2023-08-27
> > **Significant more work done in updated manuscript**
> >
> > I thank the authors for their time in considering my comments and providing an updated manuscript with significant updates and more work. This has been reflected in my updated rating. Some discussion points for future work:
> > > Contrary to some related work, we purposefully omitted anthropogenic drivers. As far as we are aware, these factors are mainly important to the ignition of fires, since human activity in various forms is often the cause of wildfires. We wanted to avoid these difficulties inherent to risk prediction and focus on predicting the spread of already existing wildfires. We do, however, provide the land cover feature, which includes “Urban and Built-up Lands” that can be seen as a binary proxy for human activity.
> >
> > Indeed this is a valid point. However fire suppression resources impact fire spread and these cannot be implicitly modeled with your dataset. For example, the road network (extracted from OSM for example) may be a proxy for the difficulty encountered by firefighters to reach a remote wildfire to constrain its spread.
> >
> > > We have also considered your advice about adding tracks and believe that it is a very good approach to provide a clearer starting point and framework for readers to get started with. We have not adopted it for this paper, because the decision of which features to include for which tasks, and which tasks might be easy or difficult are not very clear to us yet.
> >
> > This would be a nice addition for a future direction. Including tracks facilitates benchmarking in the community and follow-up research works.
> >
> > >Which information would you like to see visualized in the figure? We will add an example of how active fire detections look for several days in a row to the supplementary material to account for the second part of your request.
> >
> > I was simply referring to visually encoding what you are describing in Section 2.2.1, the layers used and how you extract the time-series: "To perform multi-temporal predictions with a fixed time window, we also need to have observations from before the first day of the fire event. Therefore, we add four days before the beginning of the fire event in GlobFire. We also add four additional days at the end."

---

### Official Review · Reviewer_ANKW · 2023-07-23
**WildfireSpreadTS**

**Rating:** 4
**Confidence:** 4

**Strengths:**

•	The authors have taken a comprehensive approach, recognizing the significance of long-term wildfire prediction. They have collected relevant data before and after the wildfires and increased the spatial resolution of the wildfire data.  This enables developers to build multi-day inputs and customize prediction tasks effectively.
•	The authors have released this dataset to encourage researchers to further investigate wildfire propagation using methods such as multi-temporal analysis, noise reduction, or generative techniques


**Additional Feedback:**

•	Wildfires are chaotic systems, and their spread is an extremely significant phenomenon. Therefore, employing appropriate methods or obtaining additional datasets to enhance the spatio-temporal resolution of the existing dataset will not only render it more meaningful but also greatly assist in studying the intricate relationship between the wildfire spread process and other influencing factors.

For the UTAE model, the number of parameters in the model is very large, and the complexity of the network increases. Training strategies such as strict regularization or early stop methods should be added to optimize the results and become a better benchmark


**Clarity:**

•	There are some symbol errors in the article, such as the period after "years" in line 18. The colon ":" in the description of Table 1 and the comma after 2018 in Table 2. Additionally, there is a missing space before the citation in line 205.
•	Why is it necessary to encode the 17 land cover types using one-hot encoding? Can't we create a map directly instead?
•	Line 59 The abbreviation “GRIDMET” should be given its full name the first time it appears


**Correctness:**

•	The data acquisition and processing methods are similar to previous studies, and the experimental design is also reasonable. But the aggregation of high and low spatio-temporal resolution worries me.

**Documentation:**

•	I believe the authors have provided sufficient details regarding data collection and organization, availability and maintenance, as well as ethical and responsible use.
•	Additionally, they have made the benchmark source code available on GitHub (https://github.com/SebastianGer/WildfireSpreadTS), allowing users to reproduce the results and get started easily.


**Ethics:**

•	There is no known ethical issue

**Limitations:**

please see the last question

**Opportunities For Improvement:**

•	I am not sure if it is an issue of model optimization or if these additional features are not needed in the wildfire spread task. To clarify this, we need to include an experiment with a ResNet18 U-Net model but without the incorporation of multi-temporal and all channels.
•	The paper should provide a detailed description of how the aggregation and processing of spatio-temporal resolution are performed for other datasets. Simple interpolation may have a negative impact on the prediction task. Why not attempt a multimodal approach using data with different spatial resolutions to demonstrate the benefits of incorporating new data?
•	As far as I know, the GRIDMET dataset also includes the burning index. Why was it not included as additional information?
•	The authors should provide a more detailed description of the differences from previous research and emphasize them. Currently, the only changes seem to be the addition of time information, higher resolution, and more features. While higher resolution indeed proves effective, many of these additional input data may not have such high resolution. Interpolation of terrain data may not have a significant impact, but certain meteorological data could have a substantial influence, and this should be explicitly stated in the paper


**Relation To Prior Work:**

•	Overall, it is clearly discussed how this work differs from previous contributions. However, it appears that some previous study has been ignored, and the author did not emphasize their own advantages enough in comparison with NextDayWildfireSpread
Singla S, Mukhopadhyay A, Wilbur M, et al. Wildfiredb: An open-source dataset connecting wildfire spread with relevant determinants[C]//Conference on Neural Information Processing Systems Track on Datasets and Benchmarks. 2021.
Sayad Y O, Mousannif H, Al Moatassime H. Predictive modeling of wildfires: A new dataset and machine learning approach[J]. Fire safety journal, 2019, 104: 130-146.
Tavakkoli Piralilou S, Einali G, Ghorbanzadeh O, et al. A Google Earth Engine approach for wildfire susceptibility prediction fusion with remote sensing data of different spatial resolutions[J]. Remote sensing, 2022, 14(3): 672.


**Summary And Contributions:**

This paper introduces a novel remote-sensing dataset named WildfireSpreadTS, designed for predicting the spread of active wildfires at a 24-hour resolution. The dataset comprises 13,607 images encompassing 607 fire events in the United States, spanning from January 2018 to October 2021. It is a multi-temporal and multi-modal dataset, providing valuable information on fuel, topography, weather conditions, and detected active fires. Such richness allows for the development of sophisticated models that can consider multiple factors influencing wildfire spread. The dataset is publicly available, serving as an important benchmark for wildfire spread prediction and facilitating the creation of new and more accurate models.

---

> ### Author Response · Authors · 2023-08-18
>
> > “The paper should provide a detailed description of how the aggregation and processing of spatio-temporal resolution are performed for other datasets. Simple interpolation may have a negative impact on the prediction task. Why not attempt a multimodal approach using data with different spatial resolutions to demonstrate the benefits of incorporating new data?”
>
> The closest dataset to ours is NextDayWildfireSpread, for which the spatial aggregation works just the way that we do it. Everything is resampled to the resolution of the target data. An approach that does not do this would probably require a custom architecture, which then makes it a lot harder for the community to try their own approaches. Our goal is to make it easy for researchers to try their methods on this problem, which is why this approach does not fit into the current paper.
> The question of how to efficiently use the knowledge that features have different resolutions has indeed been discussed amongst ourselves at the very beginning of the project. We unfortunately did not find any answers that seemed much more promising than resampling, while being accessible and keeping file size at reasonable levels.
>
> > “As far as I know, the GRIDMET dataset also includes the burning index. Why was it not included as additional information?“
>
> The ERC and drought index from GRIDMET were added in accordance with Huot et al. 's dataset that already used them. When choosing which features to add on top of those used in closely related work, we wanted to tend towards more unprocessed data like weather time series or VIIRS reflectance, to allow the neural networks to find something like their own indices during the optimization process, by combining the more raw data.
>
> > “Interpolation of terrain data may not have a significant impact, but certain meteorological data could have a substantial influence, and this should be explicitly stated in the paper”
>
> For the majority of pixels, the resampling simply consists of repeating the lower resolution value several times. The value would only get interpolated at the border between two different low-resolution pixel values. We could add an explicit statement that the values should not be taken as perfect ground truth, due to resampling. Would that address your criticism?
>
> > “The aggregation of high and low spatio-temporal resolution worries me.”
>
> We view the dataset as a way to enable computer vision researchers, with little background in remote sensing, to work on the task of wildfire spread prediction. For this, it’s very useful to have all the data in a common format and resolution, so it can be fed into existing architectures that these researchers focus on (see reviewer KxL4’s comments about the dataset not being ML-ready). The dataset should not be interpreted to indicate the perfectly correct value of each variable at each pixel, with that resolution. The resampling is a technical necessity, to make the data usable by the models that are widely used in our field today, and that require a common resolution of all channels. The alternative would be to keep all of the features separated from each other, and require researchers to build custom architectures to work with the data. This would, however, make the dataset a lot less accessible to researchers in our field of computer vision.
>
> > “Why is it necessary to encode the 17 land cover types using one-hot encoding? Can't we create a map directly instead?“
>
> The problem here is that conventional neural networks can only handle floating point numbers as input, not categorical variables like the land cover type. If we input them as integers, they will be interpreted by the neural network to be continuous variables, not categorical. For example, the difference between “Evergreen Needleleaf Forests” (index 0) and “Mixed Forests” (index 4) would be deemed larger than the difference between “Barren” (index 15) and “Water” (index 16). In reality, of course, the class indices are not ordinal, nor continuous variables, but nominal, so these distances would suggest relationships to the network that don’t exist. How to integrate categorical features into neural networks is not a solved problem. We chose one-hot encodings, others might choose to compute low-dimensional embeddings.
>
> > "It appears that some previous study has been ignored, and the author did not emphasize their own advantages enough in comparison with"  A) NextDayWildfireSpread, B) WildfireDB, C) Predictive modeling of wildfires, D) A Google Earth Engine approach [...]
>
> We made several changes in the related work section and the abstract to better emphasize the benefits of the proposed dataset compared to NextDayWildfireSpread. We have added WildfireDB in the related work section. Papers C and D both model wildfire risk and have been added to the citation list of wildfire risk datasets in the introduction. Since we model wildfire spread, not risk, we do not go into further details in the comparison.

---

### Official Review · Reviewer_hHGN · 2023-07-28
**Review of "WildfireSpreadTS: A dataset of multi-modal time series for wildfire spread prediction"**

**Rating:** 7
**Confidence:** 3
**Correctness:** The claims and the methods of the pap…
**Clarity:** Yes, the paper is well written.

**Strengths:**

- Integrating the wildfire data with other climate variables is not an easy task and it is time-consuming. Therefore, it is nice that the dataset includes other weather and climate variables like wind speed, min and max temperature, precipitation etc.
- The authors recommend (and provide code) that the data be converted to HDF5, which is a versatile, interoperable file format that is supported across the platforms and programming languages

**Additional Feedback:**

No additional feedback at this time

**Documentation:**

- The authors will share the data on Zenodo, which is good
- The GitHub repository looks alright, though more documentation would be even better
- The repository on how the datasets were created (https://github.com/SebastianGer/WildfireSpreadTSCreateDataset/tree/master) can be vastly improved with more information and step-by-step documentation. There is currently very little documentation on the code and no code comments. This is a critical part of a data product and has impacted the rating of the paper.

**Ethics:**

I don't believe that there are any ethical concerns.

**Limitations:**

I believe the authors have adequately addressed the limitations and potential negative impacts.

**Opportunities For Improvement:**

- The train-test split along the years doesn't seem to be optimal as there are years with few wildfires. The authors should consider other train-test splits ie a split across spacial dimensions
- The dataset includes years 2018-2021 which could be expanded

**Relation To Prior Work:**

The paper includes a section on related work and references a large number of related efforts

**Summary And Contributions:**

The paper introduces a new dataset, WildfireSpreadTS, that aims to predict the future spread of active fires. The dataset combines existing data products that capture various aspects of fuel, wind, topography, and humidity conditions with active fire masks indicating the fire's current position. The dataset is organized as a collection of image time series with consistent temporal and spatial resolution. The paper conducts prediction experiments and provides baseline results (using logistic regression, u-net, convLSTM, etc) for the next day's active fire map based on either one-day (mono-temporal) or five-days (multi-temporal) of observations.

---

> ### Author Response · Authors · 2023-08-18
>
> > “The train-test split along the years doesn't seem to be optimal as there are years with few wildfires. The authors should consider other train-test splits ie a split across spatial dimensions.”
>
> The point of the problematic train-test splits was raised by several reviewers. We decided to stick with a per-year split, since this is the most realistic setting in practice. After having observed years of training data on Californian wildfires, we want to be able to predict the spread of this year’s fires in California. If the data distributions change from year to year, then this is a problem our evaluation should consider. To account for your and other reviewers’ valid criticism of the splits, we now use a full 12-fold cross-validation across years to get a more realistic performance evaluation, including the effects of ‘problematic’ years in training, testing and validation sets.
>
> > “The dataset includes years 2018-2021 which could be expanded.”
>
> Expanding the dataset is definitely an option, if there is enough interest in the community. We did not yet expand it to the full possible range, because the large number of features already makes it take up quite a lot of space, which can be a practical factor that prevents adoption in the community.
>
> > “The GitHub repository looks alright, though more documentation would be even better.”
>
> Is there anything in particular that you would like to see better explained or documented for the main repository? Note that the current version does not yet include the changes that were made since the initial submission. These changes will be contributed within the next week.
>
> > "The repository on how the datasets were created can be vastly improved with more information and step-by-step documentation. There is currently very little documentation on the code and no code comments. This is a critical part of a data product and has impacted the rating of the paper."
>
> We are aiming to improve this within the next week, after having focused on the more pressing matter of improving the experiments so far. We will notify you here with a comment when this has happened, so you have a chance to determine whether the changes appropriately address your criticism.

---

### Official Review · Reviewer_n9fn · 2023-07-28
**Review of WildfireSpreadTS Benchmark**

**Rating:** 8
**Confidence:** 5
**Clarity:** The paper is well written, and clear …

**Strengths:**

The dataset improves the state-of-the-art existing training data for wildfire mapping by adding new predictors such as weather conditions and topography. Providing the data in consistent formats and enabling the user to choose different temporal windows for predictors and predictions is great. The paper very well explains the issues/gaps with existing training datasets, and address those. Authors have selected the best of available satellite and weather observations to construct the dataset.

**Additional Feedback:**

N/A

**Correctness:**

The statements made throughout the paper are correct and supported by evidence. The metrics used to evaluate the model are sound, and the dataset is constructed by understanding the physics of the problem.

**Documentation:**

Dataset was available to the reviewer, and it will be shared on GitHub after publication with a CC license. There was no Documentation per se for the dataset while the paper explains the methodology of the dataset collection/curation. I suggest including a Documentation with the final publication. The dataset will be accompanied with code for users to access it though.

**Ethics:**

No ethical concerns related to this dataset.

**Limitations:**

Limitations are discussed throughout the paper, and at the end in the "Opportunities" Section. But most of these focus on modeling limitations. I was expecting authors to discuss any issues that might arise from interpolating input weather data to the spatial resolution of the active fire maps. Going from 27km to 375m of spatial resolution can lead to significant noise in the input data and little detail is provided what methodology was used for this downscaling. This can be a source of noise in the dataset and as much as weather data is essential for this prediction problem, if the quality of this downscaling is low it might have a negative impact.

**Opportunities For Improvement:**

This is a challenging problem to solve given the significant imbalance present in the dataset. I was hoping that the authors would select better modeling strategy for the baseline models to account for this. While this noted in the "Opportunities" for future work, the current paper could have looked into this (this is not to minimize the effort that has gone to the dataset curation).

**Relation To Prior Work:**

Previous work in the literature is reviewed and gaps/differences are explained and justified.

**Summary And Contributions:**

This paper introduces a new benchmark dataset and set of baseline models for predicting active wildfire spread using temporal satellite data and auxiliary variables. This is an important application given our changing climate and the need for first responders to act quickly during wildfire events. The availability of satellite data at global scale is key for this problem, but so far there hasn't been enough research in curating a high-quality and standard benchmark to tackle this time-series modeling problem. This paper establishes a new benchmark by aggregating several input datasets which based on the physics of wildfire are relevant to predicting the spread. Overall, the dataset is well structured and the methodology is well explained.

---

> ### Author Response · Authors · 2023-08-18
>
> > “I was hoping that the authors would select better modeling strategy for the baseline models to account for [the significant imbalance present in the dataset].”
>
> Based on the feedback by several reviewers, our experiments are now based on a full 12-fold cross-validation across years. The imbalance in the data distribution between years that became visible during these cross-validation runs is also now explicitly addressed in the paper.
>
> The class imbalance is addressed by the loss functions we use, which all work well under class imbalance (Jaccard, Dice, weighted cross-entropy), as well as an oversampling of crops with active fire in the input and labels during training.
>
> If this does not address your concerns, could you give some more concrete examples which of the more advanced approaches you would have realistically liked to see in a dataset paper like this? Many of the approaches listed under "Opportunities" would likely require a larger number of experiments, assuming they do not work right away. Fitting these additional experiments and the explanation of the more advanced method into the paper, together with the introduction of the dataset, reasoning for and explanation of features, baseline experiments etc. seems challenging, given the page limit. We will of course try to include these advanced methods in future work.
>
> > “Going from 27km to 375m of spatial resolution can lead to significant noise in the input data and little detail is provided what methodology was used for this downscaling.“
>
> We added the information that the resampling is performed internally in Google Earth Engine via bilinear interpolation in the introduction part of section 2.
>
> At the beginning of the project we also had various discussions of how to deal with the large differences in spatial and temporal resolution. For example, the NDVI product is only updated every 8 days. In the end, we saw two viable options. The first is to build a custom architecture and keep all data with different temporal or spatial resolutions separated from each other. This would make it difficult for the community to apply their existing solutions, since it requires a custom architecture. The second option, which we chose, is to accept that the resampled weather data is represented at 375m even though the actual resolution is much lower, and that the NDVI features are the same for several days in a row, even though they are not in real life. At the border between two different values, interpolation artifacts might occur that actually do represent incorrect values. At the same time, this option makes the data much more accessible for existing architectures, enabling non-remote-sensing researchers to contribute to the field, which is exactly the goal that we have with this paper.

---

> > ### Comment · Reviewer_n9fn · 2023-08-30
> >
> > Thanks for addressing these points. I agree that it's best to make the resolution of all data the same so users can focus on developing models (while all use the same downscaling methodology). My point, which you have now addressed, was that the downscaling should be explained so users are aware of potential errors/uncertainties in that pre-processing step.
> > This is a great contribution, and thanks for developing such an essential dataset.

---

### Author Response · Authors · 2023-06-15
**Link to WildfireSpreadTS dataset**

Dear reviewers,

we are planning to publicly release our dataset on the Zenodo platform with the camera-ready version. During the review period, you can access the WildfireSpreadTS dataset under the following URL:

https://kth-my.sharepoint.com/:f:/g/personal/sgerard_ug_kth_se/EkChoAaBRlVCjbhux00_bL8BqMItbnyXcCAzlEf-wZLROw?e=uTtmTc

Click "Ladda ned" (Swedish for "download") to begin the ~50GB download.

If you would like to explore the data yourself, and are not familiar with the GeoTIFF format, you can use the rasterio package [1]. To get a quick impression of the data, you could download a single day's observation 2021/fire_25294691/2021-08-05.tif at:

https://kth-my.sharepoint.com/:i:/g/personal/sgerard_ug_kth_se/EfZ3QD3euexPv6IAyDj4hQUBApxt2uTcOME4qsMi0TCXyQ?e=MiY3Dg

The image preview does not make any sense, since the image has more than three channels. This is normal. You can then read the GeoTIFF image into a numpy array:


    import rasterio

    img_path = "path/to/2021-08-05.tif"
    with rasterio.open(img_path, 'r') as f:
        img = f.read()


To get a visualization like the ones used in Figure 1, you could use matplotlib:


    import matplotlib.pyplot as plt
    import numpy as np

    fig, ax = plt.subplots(3,8)
    for i, current_ax in enumerate(ax.flat):
        current_ax.axis('off')
        if i<=22:
            current_ax.imshow(np.nan_to_num(img[i,...],0.0))

Best regards

Sebastian Gerard



[1] https://rasterio.readthedocs.io/en/stable/installation.html

---

### Author Response · Authors · 2023-08-18
**Added several cross-validated ablation studies, results much clearer now, core contribution proves useful for predicting wildfire spread**

In response to the comments of several reviewers, we have made large changes to our experimental section. Since the corresponding concerns were shared by several reviewers, and the changes to the experimental section are rather large, we address them in this message to all reviewers. We will address the individual reviewers’ concerns in individual responses.

We uploaded the **updated paper as supplementary material**, with **changes highlighted** in colored text, to facilitate the review process. We will upload the finalized revision without this highlighting during the next week.

## Cross-validation reveals big differences between years
The most fundamental concern that we saw in the reviews was that the experimental results did not seem very clear and that it was difficult to tell whether this hinted at problems in the dataset aggregation, training setup or optimization. To address this, we followed the recommendations of several reviewers to add cross-validation to the evaluation. The result was that **testing on the year 2019** (as we did in the initial submission) leads to **very low scores**. Similarly,  **testing on 2021** results in rather **high scores**, while the other two years result in a medium performance. Section 6.2 in the supplementary material provides more information on this. Consequently, we now perform a **full 12-fold cross-validation for all experimental results**, to get an adequate estimate of the model performance.

## Ablation studies
Another concern raised by reviewers was that it was difficult to judge the performance of the newly added features. To address this, **we added cross-validated ablation studies for each individual feature group** (weather, topography, etc.) combined with **one to five days of input data**. A common result of all of these experiments is that adding more input observations improves the model’s performance. This clearly **validates the utility of our core contribution**: providing multi-temporal data for wildfire spread modeling.

## Baseline experiments
After switching the evaluation to the full cross-validation across years, our baselines now all clearly beat the simple persistence baseline. We also switched to using the **average precision (AP) metric for all experiments**, since it considers all possible thresholds, unlike the F1 metric that we used before. This proved especially important for the UTAE model. This change was partly in response to the reviewer who asked for AUROC plots to better judge the previously unclear performance of the models. **The results are shown in Table 5, which is also attached to this message.**

Overall, we believe that these changes, together with the smaller changes, as result of the very thorough reviews we received, make the paper much stronger and more useful to the community. We thank you for this contribution and the respect you showed us and our work by investing significant time and effort into your reviews.

|  |  |  |  |  |  |
|:---|:---:|:---:|:---:|:---:|---:|
| **Model** | **Input days** |  | **Features** |  | **Parameters ('All')** |
|  |  | **Vegetation** | **Multi** | **All** |  |
| Persistence | 1 | 0.183 ± 0.065 | 0.183 ± 0.065 | 0.183 ± 0.065 | 0 |
| Log. Regression | 1 | 0.272 ± 0.092 | 0.280 ± 0.091 | 0.280 ± 0.092 | 361 |
| Res18 U-Net | 1 | 0.309 ± 0.092 | 0.325 ± 0.090 | 0.312 ± 0.105 | 14.4M |
| Res18 U-Net | 5 | 0.324 ± 0.086 | 0.333 ± 0.081 | 0.301 ± 0.097 | 14.7M |
| ConvLSTM | 5 | 0.306 ± 0.082 | 0.310 ± 0.085 | 0.292 ± 0.094 | 240K |
| UTAE | 5 | **0.370 ± 0.104** | **0.355 ± 0.121** | **0.337 ± 0.128** | 1.1M |

---

### Author Response · Authors · 2023-08-26
**Final revision: Adjusted evaluation, added the promised small changes.**

Dear reviewers,

we have now uploaded our final revision, including all of the smaller changes that we promised in our last comments.

Furthermore, we had discovered a small error in our evaluation pipeline, that meant that the test scores were not perfectly comparable across different amounts of leading days. We re-ran the corresponding experiments, so runs using one day's observation are now evaluated on precisely the same test sets as runs using five days. See section 5.2, second paragraph for the details. This adjustment increased the performance by a few percentage points for many runs. While some details changed, the general trends in the experimental results stayed the same.

The small changes we made to address reviewers' concerns are:

- Adding an example of how the active fire detections evolve over time, for one interesting fire, in the supplementary material, section 5.2
- Adding statistics on how fire/no-fire pixels evolve from day to day in the supplementary material, section 5.3
- Adding the amount of fire and missing data in the supplementary material, section 5.4
- Reworking Figure 3 in the main paper to be more easily comprehensible
- Adding more information in section 5.2 on how the data used in the experiments is extracted from the full dataset
- Adding a comment to the datasheet in the supplementary material, section 1.5,  emphasizing that the original resolution of features vary, even though they were all resampled to a common resolution, and that the dataset does not represent accurate measurements at the resampled resolution
- Adding further code comments and doc strings to the main code repository: https://github.com/SebastianGer/WildfireSpreadTS
- Cleaning up the code of the dataset creation repository, adding doc strings and code comments: https://github.com/SebastianGer/WildfireSpreadTSCreateDataset

We want to again thank you for the time that you invested into reviewing our work and helping us to turn it into a much stronger paper.

---

### Decision · Program_Chairs · 2023-09-22

**Decision:**

Accept (Poster)

**Comment:**

* Summary. The manuscript introduces a valuable multi-temporal, multi-modal dataset for predicting wildfire spread. It contains 13,607 images from 607 fire events, providing a rich resource for research. The dataset's time series structure enables in-depth analysis of wildfire dynamics and includes diverse environmental variables. Challenges include multi-temporal inputs, many input channels, imbalanced and noisy labels, and the complexity of physical processes. Despite these challenges, the dataset offers opportunities for advancing wildfire prediction methods and encouraging research in noise-resistant and generative techniques.
* On reviewer’s comments. Reviewers' opinions on the paper vary. Reviewer 1 rates it 8 and strongly recommends acceptance, lauding the new wildfire prediction dataset and suggesting improvements. Reviewer 2 gives it a 7 and recommends acceptance, appreciating the climate variables but pointing out the need for better documentation and dataset expansion. Reviewer 3 rates it 4 and suggests rejection due to concerns about additional features and data resolution. Reviewer 4, with a confidence rating of 5, strongly recommends acceptance (rating 7) and praises the dataset's significance. However, it offers several suggestions for improvement, including making it ML-ready and conducting experiments without reflectance data. Reviewer 5 initially rates it marginally above acceptance (6) but raises their rating after revisions, expressing concerns about mixed baseline results and dataset design. They stress the need to demonstrate the significance of new features and suggest alternative evaluation setups for reproducibility. Overall, the paper obtains quite uneven ratings and feedback, with some reviewers highlighting its importance and others offering substantial improvements and clarifications.
* Author’s responses. The authors engaged in a fruitful and constructive discussion and provided convincing explanations to the reviewer’s comments, addressing most of their concerns. After the multiple review comments, the general feeling is that most problems have been discussed competently, and one reviewer even upgraded the initial rating.
* On this work's quality, clarity, originality, and significance. The work is of high quality, offering a valuable wildfire prediction dataset. However, clarity could be improved, and some reviewers noted concerns about dataset features and baseline results. While the paper's originality lies in its dataset, addressing these concerns is essential for its significance to the wildfire research community.